# Introducing dorsoventral patterning in adult regenerating lizard tails with gene-edited embryonic neural stem cells

Thomas P. Lozito [1,2 ✉], Ricardo Londono[3], Aaron X. Sun[3] & Megan L. Hudnall[1]

Lizards regenerate amputated tails but fail to recapitulate the dorsoventral patterning achieved during embryonic development. Regenerated lizard tails form ependymal tubes (ETs) that, like embryonic tail neural tubes (NTs), induce cartilage differentiation in surrounding cells via sonic hedgehog (Shh) signaling. However, adult ETs lack characteristically roof plate-associated structures and express Shh throughout their circumferences, resulting in the formation of unpatterned cartilage tubes. Both NTs and ETs contain neural stem cells (NSCs), but only embryonic NSC populations differentiate into roof plate identities when protected from endogenous Hedgehog signaling. NSCs were isolated from parthenogenetic lizard embryos, rendered unresponsive to Hedgehog signaling via CRISPR/Cas9 gene knockout of *smoothened* (*Smo*), and implanted back into clonally-identical adults to regulate tail regeneration. Here we report that *Smo* knockout embryonic NSCs oppose cartilage formation when engrafted to adult ETs, representing an important milestone in the creation of regenerated lizard tails with dorsoventrally patterned skeletal tissues.

[1] Department of Orthopaedic Surgery, University of Southern California, Los Angeles, CA, USA. [2] Department of Stem Cell Biology and Regenerative Medicine, University of Southern California, Los Angeles, CA, USA. [3] Center for Cellular and Molecular Engineering, Department of Orthopaedic Surgery, University of Pittsburgh School of Medicine, Pittsburgh, PA, USA. ✉email: lozito@usc.edu

V ertebrate regeneration is often studied through the lens of embryonic development, and many of the most intensely studied models of adult appendage regeneration closely follow embryonic programs to replace lost tissues[1]. For example, during both appendage development and regeneration, specialized structures act as signaling centers that direct differentiation of surrounding tissues. During embryonic development, tail tissues form from tail buds, collections of mesodermal cells that respond to patterning signals generated by secondary neural tubes (NTs)[2–4]. Tail NTs exhibit distinct molecular differences between dorsal roof plate domains and ventral floor plate domains[5]. Roof plates express distinct sets of markers, including Pax7, Msx2, BMPs, and Wnts, while floor plates express Shh and FoxA2. Intermediate zones between roof and floor plates are referred to as lateral domains and are marked by Pax6 expression. This separation of distinct signaling molecules into dorsal and ventral domains has direct consequences on dorsoventral patterning of a range of tissues, including axial skeletons. For example, Shh produced by NT floor plates induce skeletogenesis in tail bud notochords and sclerotome but are antagonized by BMP signals from roof plates, thereby limiting early skeletal formation to ventral regions[6]. Early embryonic skeletons are made up of cartilage marked by expression of collagen type II (Col2) and are later replaced by bone[7]. Tail NTs, themselves, are rich in neural stem cell (NSC) populations which undergo neurogenesis to yield neurons of spinal cords and dorsal root ganglia (DRG)[8,9]. Remnants of embryonic NT NSCs persist into adulthood among ependymal/radial glial cell populations lining central canals of tail spinal cords[10]. Tail regeneration begins with the growth of blastemas on amputated tail stumps[11–13]. These collections of mesenchymal cells differentiate into the majority of regenerated tail tissues, including skeletons[12,13]. Instead of NTs, ependymal tubes (ETs) sprout from original tail spinal cord ependymal populations and invade tail blastemas[12–14]. Like embryonic tail NTs, regenerated tail ETs induce differentiation in surrounding cells and contain populations of NSCs[12–14]. However, the extent to which adult and embryonic tail NSC populations resemble one another varies with species and appears to affect tail regenerative fidelity[14].

Lizards are the only amniotes capable of tail regeneration, making them the closest relatives to mammals capable of regrowing an amputated appendage[11]. Lizard tail regeneration also presents a unique case in which developmental and regenerative outcomes diverge greatly, particularly as they relate to skeletal tissue patterning[13]. Tail vertebral columns that form during embryonic development are regenerated as unpatterned and unsegmented cartilage tubes. Similarly, regenerated lizard tail ETs lack the dorsoventrally patterned roof, lateral, and floor domains developed by embryonic NTs[14]. We hypothesize that the lack of dorsoventral patterning in the regenerated lizard tail ETs and skeletons are directly related, and that introduction of NSC populations with the ability to dorsalize in regenerated lizard tail environments will induce patterning in both tissues. The goals of this study include comparing the differentiation potentials of embryonic and regenerated tail NSCs and the generation of dorsoventrally patterned regenerated lizard tails.

Here, we show that embryonic NSCs transplanted into regenerated tail ETs retain the capacity to form roof domains but are ultimately ventralized by the unchecked Hedgehog signaling of adult lizard tail environments. Embryonic lizard NSC lines unresponsive to Hedgehog stimulation are generated through the use of CRISPR/Cas9 technologies to knockout (KO) the signaling regulator smoothened (Smo). Exogenous Smo KO NSCs injected into adult tail spinal cords engraft to endogenous ependymal cell populations and contribute to dorsal domains in regenerated tail ETs. Embryonic Smo KO NSCs maintain roof plate identities

in vivo, and lizards treated with edited NSCs regrew tails that lacked cartilage in dorsal regions.

## Results

**Regenerated lizard tails lack dorsoventral patterning.** Embryonic lizard tails begin as tail buds, collections of mesodermal cells surrounding neural tubes (NTs) populated by Sox2[+] neural stem cells (NSCs) (Supplementary Fig. 1). Mesodermal cells respond to signals from embryonic NSCs to proliferate and differentiate into tail tissues. Specifically, Shh produced by tail NT NSCs induce cartilage differentiation in embryonic sclerotome from which tail skeletons are derived[6]. Similarly, regenerated lizard tails begin as blastemas, specialized regenerative structures made up of heterogenous fibroblastic cell populations surrounding central ETs (Supplementary Fig. 1)[13]. Like embryonic NTs, regenerated tail ETs contain populations of Sox2[+] NSCs that act as signaling centers for regulating patterning in surrounding tissues (Supplementary Fig. 1)[14]. We have previously shown that lizard tail blastema cells respond to proliferation and differentiation signals produced by ET NSCs[13]. Specifically, Shh secreted by ET NSCs induce blastema cell cartilage differentiation, essentially recreating the signaling environment responsible for embryonic tail skeletal development. Similarly, both tail bud and blastema NSCs proliferate during tail development and regeneration, respectively, and elongate with growing tails (Supplementary Fig. 1).

Despite these early similarities between embryonic and regenerated tails, developmental outcomes are very different, particularly as they relate to dorsoventral patterning of central nervous system (CNS) and skeletal tissues (Fig. 1). Embryonic tails exhibit NTs dorsal to notochords (Fig. 1A). Sox2[+] NSCs populate NTs and undergo neurogenesis to differentiate into Tuj1[+] neurons of early spinal cords and DRG (Fig. 1A)[8,9]. Meanwhile, notochords and surrounding tissues express Col2, a marker of the early cartilaginous skeleton (Fig. 1A). This pattern of dorsal CNS and ventral skeletal tissue is maintained through embryonic development and into adulthood (Fig. 1B). Neural tube NSCs differentiate and add more neurons to spinal cords and DRG (Fig. 1B), while notochords and surrounding sclerotome develop into the vertebrae of adult tails (Fig. 1B). Sox2[+] NSCs persist in adult spinal cords among ependymal cell populations lining central canals, the remnants of embryonic NTs (Fig. 1B). Upon amputation, adult lizard tails regenerate, but dorsoventral tissue patterning is replaced by radial symmetry around central ETs (Fig. 1C). ETs are derived from adult tail spinal cord ependyma populations and are enriched for Sox2[+] NSCs (Fig. 1C). Cartilage tubes form from blastema cells and completely surround ETs (Fig. 1C). Spinal cord neurons and DRG are not regenerated in regrown tails (Fig. 1C).

The loss of regenerated tail CNS and skeletal tissue patterning is preceded by the disappearance of patterned NSC populations during NT maturation (Fig. 1D–O). Embryonic tail NTs exhibit distinct molecular differences between dorsal roof plates, ventral floor plates, and intermediate lateral domains. NT roof plates express the markers Pax7 and BMP4 (Fig. 1D, E); lateral domains express Pax6 (Fig. 1F); and floor plates express Shh and FoxA2 (Fig. 1D, F, G). Interestingly, lizard tail NT Pax6 expression exhibited a dorsal shift in Pax6 expression compared to what has been reported for amniote trunk NTs[15,16], and may represent a difference between secondary vs primary NTs. Adult tail spinal cord ependyma, the direct descendants of embryonic NT cells, lack roof and lateral domain markers (Fig. 1H–J) but retain floor plate markers Shh and FoxA2 (Fig. 1H, J, K). Shh protein was also detected among spinal cord nerves surrounding original tail ependyma. In turn, regenerated tail ETs cells are derived from

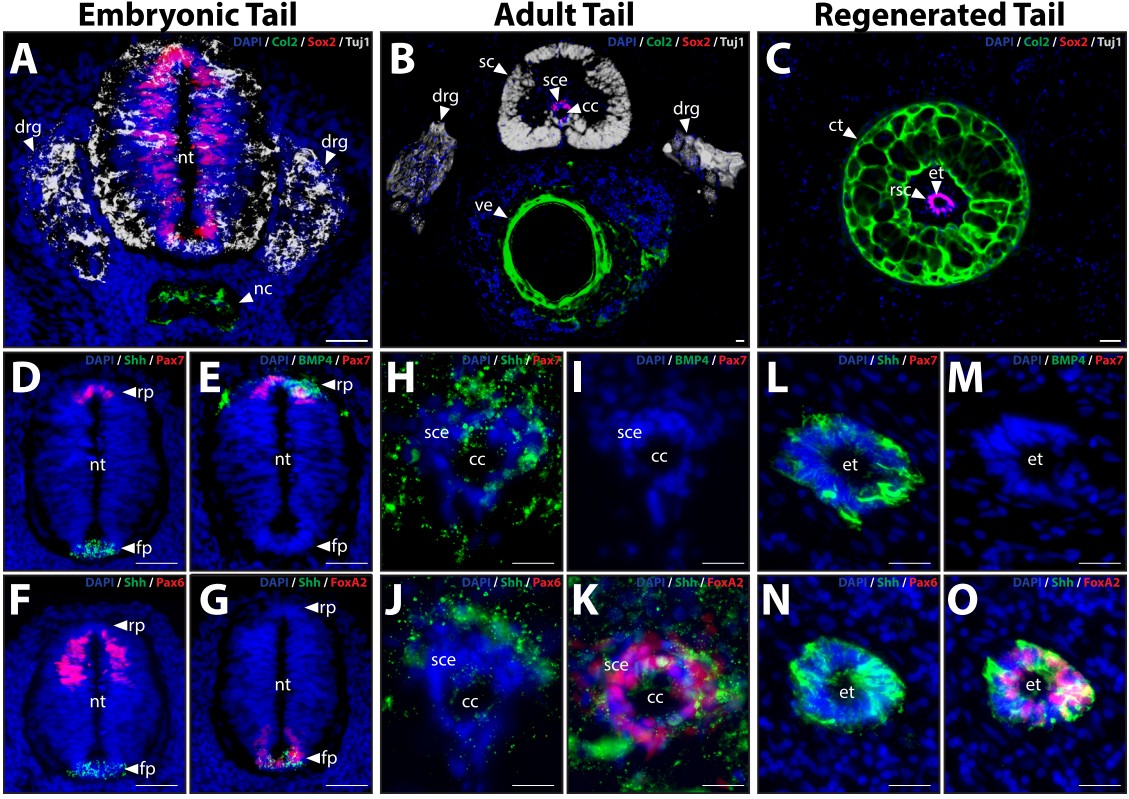

**Fig. 1 Dorsoventral patterning of skeletal and CNS tissues established during embryogenesis is not recapitulated during lizard tail regeneration.** Cross sections of **A** embryonic (14 days post-oviposition; DPO), **B** adult (28 days post hatching; DPH), and **C** regenerated lizard tails (28 days post-amputation; DPA) were analyzed by Col2, Sox2, and Tuj1 IF to highlight the spatial arrangements of skeletal (cartilage), NSC, and nerve tissue, respectively, during tail development and regrowth. **D–O** Cross sections of NT and spinal cord ependymal structures were analyzed by Pax7, Pax6, Shh, and FoxA2 IF to compare the expression of roof plate, lateral domain, and floor plate identities among **D–G** embryonic, **H–K** adult, and **L–O** regenerated tails. Ten tails belonging to each developmental/regenerative stage were analyzed. cc central canal, ct cartilage tube, DPA days post-amputation, DPO days post-oviposition, DPH days post hatching, drg dorsal root ganglion, et ependymal tube, fp floor plate, nt neural tube, rp roof plate, rsc regenerated spinal cord, sc spinal cord, sce spinal cord ependyma. Bar = 50 μm.

original tail ependyma cell populations and also lack roof and lateral domain identities (Fig. 1L–N), being entirely made up of Shh$^+$ Foxa2$^+$ floor plate (Fig. 1L, N, O). Taken together, these results demonstrate that dorsoventral patterning of skeletal and CNS tissues developed during embryogenesis are not recapitulated during lizard tail regeneration. We hypothesize that these findings are related; regenerated lizard tail skeletal tissues lack dorsoventral patterning because ETs lack patterned NSC populations. Patterning of embryonic NT NSCs is lost during adult tail ependymal cell derivation, which in turn results in unpatterned regenerated tail ETs and cartilage tubes.

Interestingly, we attempted to test the regenerative potential of embryonic lizard tails, but results have been inconclusive. Embryonic tails amputated in ovo (7 and 14 DPO) failed to regenerate. While embryos remained viable for weeks following surgery and egg re-closure, amputated tail stumps became bound to amnion membranes, which may have interfered with tail healing. Tails of embryos removed from eggs and cultured in vitro also failed to regrow following amputation, but this may have been an artifact of prolonged culture conditions.

**Embryonic and adult lizard NSCs differ in dorsoventral patterning.** Since Sox2$^+$ NSCs were detected in embryonic and adult tail tissues, we sought to characterize and compare the roof plate, lateral domain, and floor plate identities of these cells in vivo and in vitro (Fig. 2). We have previously shown that adult lizard tail NSCs form neurospheres in response to FGF stimulation in

culture (Fig. 2C, D)[14], and we tested whether this held true for embryonic tail NSCs. Both embryonic tail NTs and adult tail spinal cords were isolated via microdissection, digested with papain, and cleared of myelin to yield single-cell suspensions (Fig. 2A, C). Both embryonic NT and spinal cord NSCs formed neurospheres after 14 days in culture in the presence of FGF (Fig. 2B, D). Next, Pax7, Pax6, and Shh expression were analyzed to determine the positional identity of embryonic vs. adult tail Sox2$^+$ NSCs in situ and after neurosphere formation in vitro (Fig. 2E–P). Sox2$^+$ cells were detected among Pax7$^+$ roof plates, Pax6$^+$ lateral domain, and Shh$^+$ floor plates of embryonic tail NTs (Fig. 2E–G). In contrast, adult lizard tail spinal cord ependyma and their resident Sox2$^+$ NSCs expressed Shh only (Fig. 2H–J). In vitro, both embryonic NT- and adult spinal cord-derived neurospheres were enriched for Sox2$^+$ NSCs (Fig. 2K–P). Embryonic neurospheres were predominantly Pax7$^+$, Pax6$^-$, Shh$^-$ (Fig. 2K–M) (Supplementary Fig. 2), while adult neurospheres were Pax7$^-$, Pax6$^-$, and Shh$^+$ (Fig. 2N–P) (Supplementary Fig. 2). These results indicated a divergence in roof plate vs floor plate identities between NT and adult spinal cord NSCs when cultured in vitro; embryonic NSC neurospheres defaulted to a roof plate identity, while adult tail neurospheres were restricted to floor plate.

**Hedgehog signaling patterns embryonic, but not adult, lizard NSCs.** Previous studies have demonstrated the importance of Hedgehog signaling in regulating cell differentiation and patterning in both embryonic and regenerated tissues[12–14]. Given the

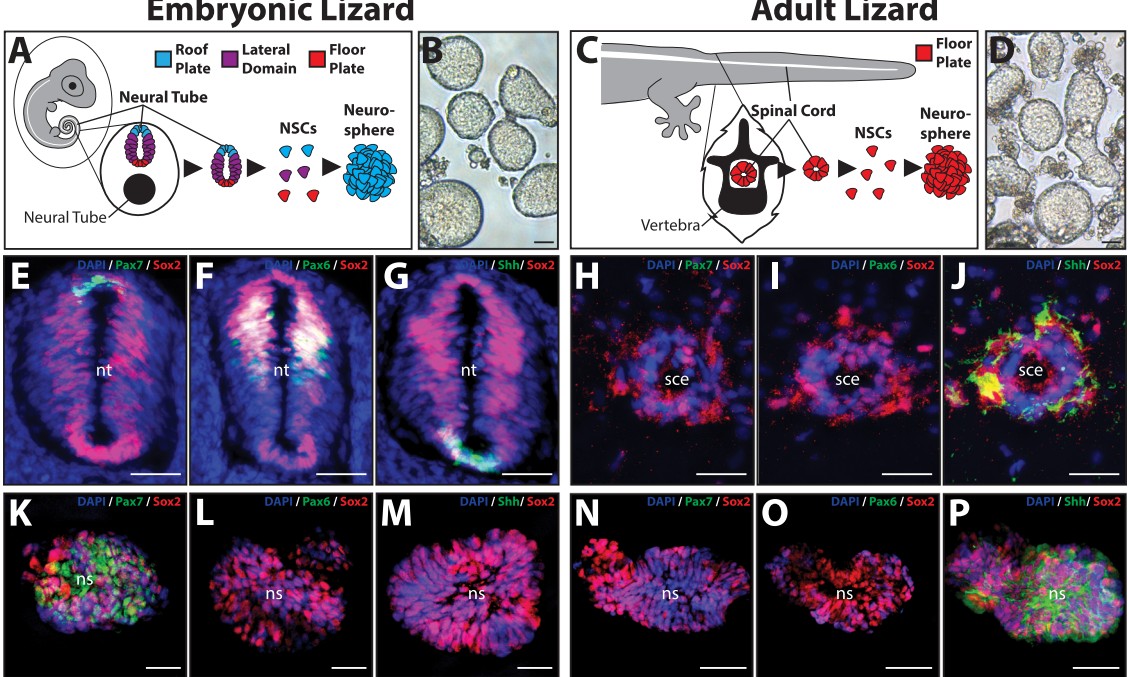

**Fig. 2 Embryonic lizard tail NTs and adult lizard spinal cord ependyma both contain NSC populations but differ in dorsoventral positional identities.**
**A**–**D** Summary schematic of embryonic tail NT and adult tail spinal cord NSC isolation and culture as neurospheres. Neurospheres derived from embryonic tails default to roof plate identity, while neurospheres derived from adult tails persist as floor plate identity. **B**, **D** Light microscopy images of 14-day-old neurospheres formed in vitro from cells isolated from **B** embryonic NTs and **D** adult spinal cords. **E**–**J** Cross sections of **E**–**G** embryonic tail NTs and **H**–**J** adult tail spinal cord ependyma analyzed by IF for roof plate (Pax7), lateral domain (Pax6), and floor plate (Shh) markers. Sox2 IF marks resident NSC populations. **K**–**P** Roof plate, lateral domain, and floor plate IF analysis of Sox2$^+$ neurospheres derived from either **K**–**M** embryonic or **N**–**P** adult tails and cultured for 14 days in vitro. Note the absence of lateral and floor plate markers in embryonic neurospheres contrasted to the lack of lateral and roof plate markers in adult neurospheres. Six tails and 250 neurospheres belonging to each developmental stage were analyzed. ns neurosphere, nt neural tube, sce spinal cord ependyma. Bar = 50 μm.

differences in skeletal patterning and NSC identity among embryonic and regenerated lizard tails, we hypothesized that divergent Hedgehog signaling is responsible for disparities in cell differentiation during tail development and regrowth. The responses of embryonic vs adult regenerated tails to inhibition and exogenous activation of Hedgehog signaling were compared (Fig. 3A–L). To treat embryonic lizard tails, freshly laid lizard eggs ($n = 6$) were injected with either the Hedgehog inhibitor cyclopamine or the Hedgehog agonist SAG and allowed to develop for one week. To treat regenerated lizard tails, cyclopamine or SAG was administered to lizards ($n = 6$) for four weeks following tail amputation. Embryonic and regenerated tail samples were analyzed for Col2 and Tuj1 expression as markers for cartilage and nerve differentiation, respectively, and for Pax7 and Shh expression as indicators of dorsoventral patterning (Fig. 3A–L). Lizard embryos treated with vehicle control developed patterned tails with Tuj1$^+$ NTs and DRG in the dorsum and Col2$^+$ notochords in the ventrum (Fig. 3A). Control embryonic tail NTs expressed Pax7$^+$ dorsal roof plate and Shh$^+$ ventral floor plate (Fig. 3B). Adult lizards treated with vehicle control regenerated tails with characteristic Col2$^+$ cartilage tubes surrounding ETs exhibiting circumferential Shh expression without Pax7 expression (Fig. 3C, D). No Tuj1$^+$ neurons or DRG were detected (Fig. 3C, D). Treatment with cyclopamine inhibited embryonic lizard tail Col2 expression and resulted in tails without noticeable notochords (Fig. 3E). Conversely, cyclopamine treatment increased embryonic tail Tuj1 expression and resulted in ectopic nerve differentiation throughout tail tissues (Fig. 3E). Tail NTs of embryos treated with cyclopamine exhibited abnormal shapes with ventral openings and marked enhancements in Pax7

expression and reductions in Shh signal (Fig. 3F), indicating dorsalization of NT phenotypes. Like embryonic tail cartilage, regenerated lizard tail cartilage tube formation was inhibited by cyclopamine treatment (Fig. 3G). However, unlike the observations of embryonic tails, cyclopamine did not affect regenerated lizard tail Tuj1 expression, and DRG remained absent (Fig. 3G). ETs remained positive for Shh expression, and Pax7 levels remained undetectable (Fig. 3H). SAG treatment induced ectopic Col2$^+$ cartilage formation in both embryonic and regenerated lizard tails (Fig. 3I, K). The response of regenerated lizard tails to SAG treatment was particularly strong, with extensive cartilage infiltration into various tail regions (Fig. 3K). In embryonic tails, SAG treatment resulted in the replacement of Pax7 with Shh expression, reductions of roof plate domains, and expansions of floor plate regions, suggesting ventralization of embryonic tail NTs (Fig. 3J). Conversely, regenerated lizard tail ETs did not respond to SAG treatment; Shh expression was maintained by the entire tube, and Pax7 expression remained undetectable (Fig. 3L). Interestingly, the replacement of Pax7$^+$ roof plate with Shh$^+$ floor plate in tail NTs of SAG-treated embryos bore striking resemblances to regenerated lizard tail ETs, suggesting that tail environments saturated with Hedgehog signaling destroyed dorsoventral patterning of NSC structures. Taken together, these results suggested patterning of both embryonic and regenerated tail cartilage was regulated by Hedgehog signaling. However, only embryonic tail NTs, and not regenerated tail ETs, responded to Hedgehog dorsoventral patterning signals. Furthermore, the strong response of regenerated lizard tail cartilage to both cyclopamine and SAG treatments validated the effectiveness of drug treatment methods in modulating cell differentiation

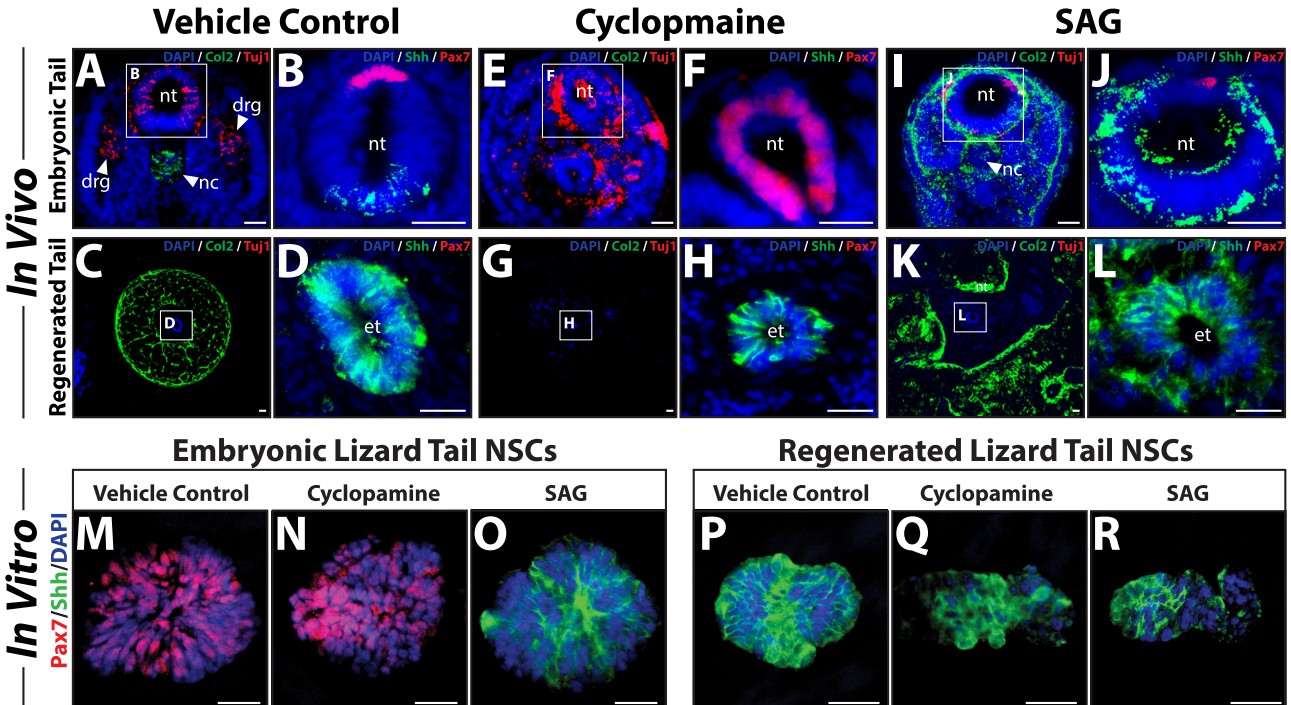

**Fig. 3 Hedgehog signaling regulates dorsoventral patterning of embryonic lizard tail NSCs but not regenerated tail NSCs, in vivo and in vitro. A–L** Cross sections of embryonic and regenerated lizard tails following treatment with vehicle control, cyclopamine, or SAG were analyzed by Col2 and Tuj1 or Pax7 and Shh IF. Neural and ependymal tube regions enclosed in white boxes are shown in magnified views to the right of the corresponding image. **M–R** Neurospheres derived from embryonic tail NTs or regenerated tail ETs were treated with vehicle control, cyclopamine (5 μM), or SAG (100 nM) and analyzed for roof (Pax7) and floor (Shh) plate identities via IF. Ten tails and 250 neurospheres belonging to each developmental stage and treatment condition were analyzed. c cartilage, et ependymal tube, nt neural tube. Bar = 50 μm.

responses to Hedgehog signaling in regenerated lizard tails and highlighted the contrasting responses of cartilage vs ET cells.

We hypothesized that differences observed between embryonic tail NT vs regenerated tail ET cell responses to modulators of Hedgehog signaling extend to neurosphere cultures of these same cell populations (Fig. 3M–R). Neurospheres were generated from either embryonic or regenerated tail NSCs and treated with either cyclopamine (5 μM), SAG (100 nM), or vehicle controls for 14 days. Neurospheres generated from embryonic tail NTs exhibited roof plate identities under control conditions (Fig. 3M) (Supplementary Fig. 3) that were unaffected by cyclopamine treatment (Fig. 3N) (Supplementary Fig. 3). In contrast, SAG treatment of embryonic NSC neurospheres induced the replacement of Pax7 with Shh expression, indicating conversion to floor plate identity (Fig. 3O) (Supplementary Fig. 3). Neurospheres derived from regenerated tail ETs were unresponsive to the same concentrations of cyclopamine and SAG (Fig. 3P–R) (Supplementary Fig. 3), and consistently expressed Shh and not Pax7. These results supported in vivo findings, where NT NSCs ventralized in response to Hedgehog activation to match the immutable floor plate identity of ET NSCs.

**Embryonic, but not adult, lizard NSCs undergo Hedgehog-influenced neurogenesis.** New neurons are formed during embryonic lizard tail development, but not adult tail regeneration[14]. To demonstrate this, embryonic, adult, and regenerated tails were analyzed for Sox2, Tuj1, and NeuN expression (Supplementary Fig. 4). NeuN is reported to be expressed in NSCs undergoing neurogenesis as well as mature neurons[17]. NeuN signal was detected in nuclei of Tuj1+ neurons in interior regions of original tail spinal cords (Supplementary Fig. 4B), but not in Sox2+ ependymal cells. NeuN was not

detected in regenerated tail spinal cord (Supplementary Fig. 4C), corroborating observations that no new neurons are formed during tail regrowth. Sox2/NeuN co-expression was only observed in embryonic tail NT cells, indicating spinal cord neurogenesis was restricted to embryonic stages (Supplementary Fig. 4A). We hypothesized that embryonic lizard tail NSCs are the sources of adult and regenerated tail neural lineages. The differentiation potentials of embryonic lizard tail NSCs were investigated in vivo by tracing the cell fates and lineage contributions of Sox2+ NT cells through tail development and into adult tail regeneration using a dual-viral vector system (see Supplementary Methods for viral system description and experimental setup and Supplementary Figs. 5–12 for presentation of results). These studies confirmed embryonic lizard NSCs contributed to CNS cell lineages of embryonic, adult, and regenerated tails.

The above experiments suggested that embryonic tail NSCs differentiated into neurons of the CNS in vivo. Here we directly compared the differentiation potential of embryonic NT and adult spinal cord NSCs in vitro. We also tested the effects of Hedgehog inhibition and activation on the neural differentiation of these cells to gain insight into the effects of the regenerated tail Hedgehog signaling environment on NSC differentiation. Embryonic NT cells and adult spinal cord ependymal cells were cultured as neurospheres and subsequently exposed to differentiation conditions for 14 days in the presence of cyclopamine, SAG, or vehicle control (Fig. 4). Prior to differentiation, both embryonic and adult NSCs expressed Sox2 and glial fibrillary acidic protein (GFAP) in neurosphere culture, and neither expressed early neurogenesis markers doublecortin (DCX) and NeuroD2 or late differentiation marker Tuj1 (Fig. 4A, B) (Supplementary Fig. 13). After differentiation, only embryonic NSCs expressed DCX, NeuroD2, and Tuj1, indicating neurogenesis in embryonic, but not adult, NSCs (Fig. 4C, D)

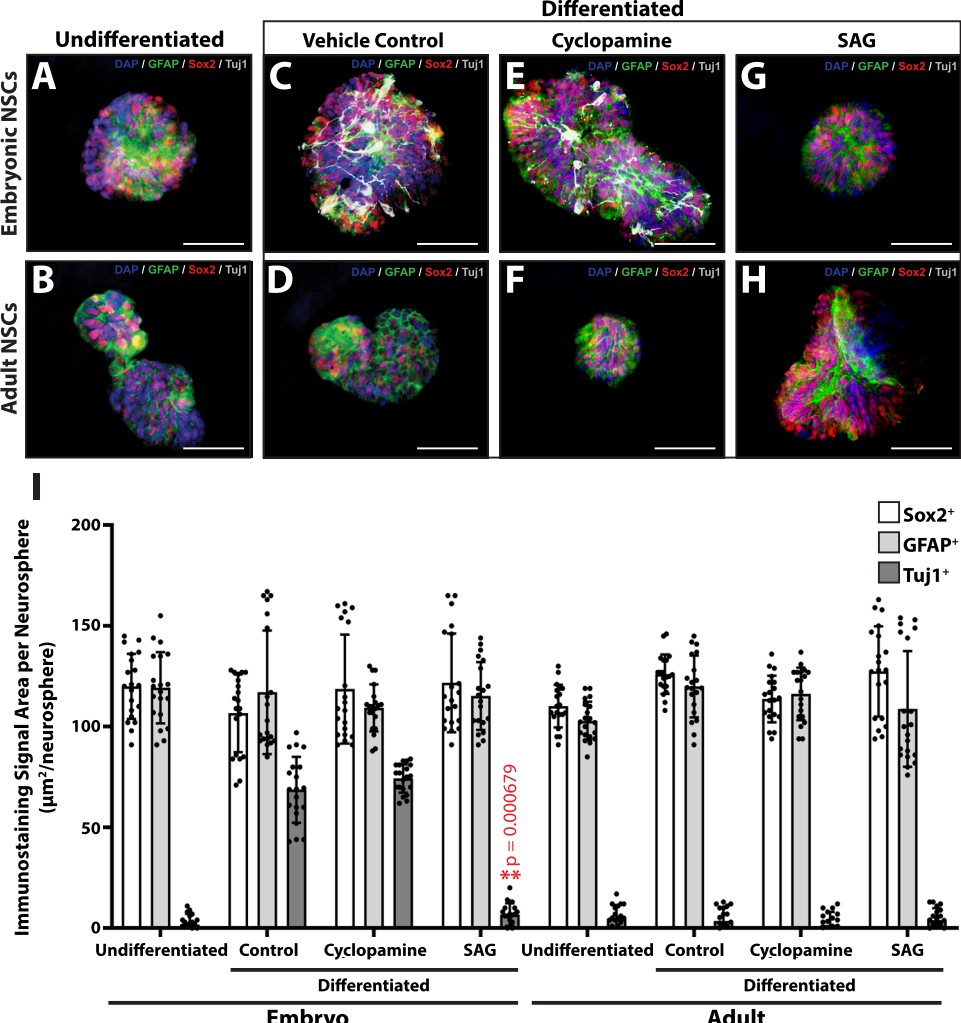

**Fig. 4 Comparisons of differentiation capacities of embryonic vs adult lizard tail NSCs and sensitivities to Hedgehog signaling in vitro.**
**A**, **B** Representative undifferentiated neurospheres derived from **A** embryonic lizard tail NTs and **B** adult tail spinal cords analyzed by Sox2, GFAP, and Tuj1 IF. **C–H** Representative embryonic and adult neurosphere cultured under differentiation conditions for 14 days treated with **C**, **D** vehicle control, **E**, **F** cyclopamine, or **G**, **H** SAG analyzed for Sox2, GFAP, and Tuj1 expression. Bar = 50 μm. **I** Quantification of embryonic and adult NSC neurosphere differentiation in response to treatment with cyclopamine and SAG. NSCs derived from embryonic and adult lizard tails were cultured as neurospheres under differentiation conditions supplemented with vehicle control, cyclopamine, or SAG and analyzed by Sox2, GFAP, and Tuj1 IF. Sox2, GFAP, and Tuj1 signal areas are presented (n = 20 for each condition). Data are presented as mean values +/− SD. Two-way ANOVA with pairwise Tukey's multiple comparison tests was used. **p < 0.001, compared to corresponding vehicle controls. Source data are provided as a Source data file.

(Supplementary Fig. 13). Embryonic NSC neurogenesis was unaffected by cyclopamine treatment but was inhibited by SAG (Fig. 4E, F) (Supplementary Fig. 13). Adult NSCs did not express DCX, NeuroD2, or Tuj1 under any condition tested (Fig. 4D, F, H) (Supplementary Fig. 13). These results suggested that embryonic, but not adult, lizard NSCs were capable of neural differentiation under Hedgehog-free conditions. Findings pertaining to the differences between embryonic and adult NSC identity and differentiation are summarized in Fig. 5.

**Embryonic NSCs transplanted into adult ETs lose roof plate identity.** We have previously shown that exogenous NSCs implanted into tail spinal cord ependyma populations contribute to regenerated ETs and maintain spatial arrangements reflecting their initial positions at injection[14]. For example, labeled cells injected into dorsal regions of original tail spinal cord ependyma proliferated and contributed to dorsal, but not ventral, regions of regenerated tail ETs. Furthermore, the differentiation of exogenous NSC populations into neural lineages within regenerated

tails was shown to be influenced by native Hedgehog signaling environments[14]. We adapted this system for use with embryonic and adult tail NSCs combined with systemic cyclopamine treatment to directly test the effects of Hedgehog signaling on NSC differentiation in vivo. NSCs were isolated from embryonic NTs and adult tail spinal cords, cultured and proliferated as neurospheres, dissociated into single cells, and labeled with the fluorescent dye DiI (Fig. 6A). DiI-labeled NSCs were implanted into dorsal spinal cord ependymal regions of amputated tails (Fig. 6A). Following 28 days of tail regrowth and treatment with vehicle control or cyclopamine, regenerated tails were collected and analyzed for DiI co-localizations with neural differentiation markers Sox2, Tuj1, and GFAP and dorsoventral patterning markers Pax7 and Shh (Fig. 6B–D). Exogenous embryonic and adult NSC populations contributed to regenerated tail ETs and maintained positional localization to dorsal regions following implantation into adult tail ependyma (Supplementary Fig. 14). Similarly, both embryonic and adult NSCs retained their Sox2+ GFAP+ glial/ependymal cell identity within regenerated tail ETs

## Tail Development

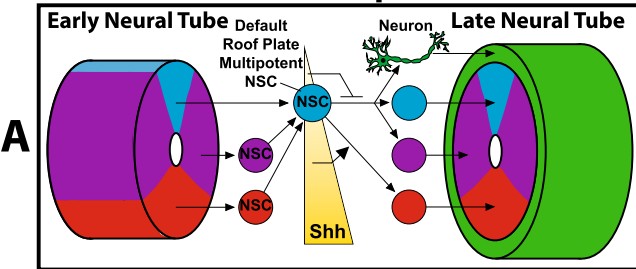

## Tail Maturation

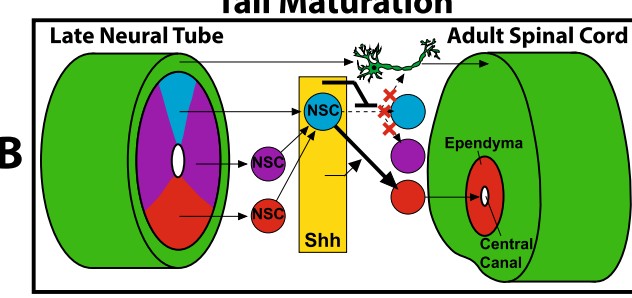

## Tail Regeneration

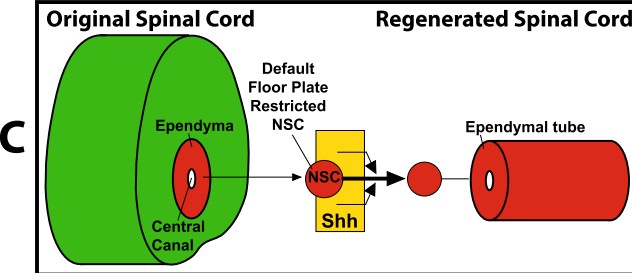

**Fig. 5 Summary of divergent NSC identities and differentiation capabilities exhibited during lizard tail development, maturation, and regeneration. A** Roof, lateral, and floor plate domains of embryonic neural tubes harbor multipotent NSCs that default to a roof plate identity during culture in vitro. Shh specifically produced by neural tube floor plates establishes Hedgehog signaling gradients that influence embryonic NSC differentiation and neurogenesis. During tail development, neural tube NSCs ventralize and form neurons according to hedgehog gradients.
**B** During tail maturation, embryonic neural tubes give rise to adult spinal cords, and remnant NSC populations persist in ependymal linings of central canals. Hedgehog gradients breaks down, inhibiting roof and lateral domain differentiation, and adult NSCs exhibit floor plate identities only.
**C** Regenerated tail spinal cords consist of ependymal tubes made up of NSCs derived from original tail ependyma. Because adult NSCs are incapable of roof/lateral domain differentiation and neurogenesis, ependymal tube NSCs are restricted to floor plate identities, and no new spinal cords neurons are formed during tail regrowth. Adult NSCs retain floor plate identities in vivo and in vitro regardless of Hedgehog singling, resulting in unregulated Shh expression and lack of Hedgehog-induced tissue patterning in regenerated tails.

and did not differentiate into Tuj1$^+$ neurons under control conditions (Fig. 6B–D). Furthermore, ET regions derived from exogenous embryonic NSCs did not express Pax7 as observed in neurosphere culture (Fig. 6B, D). Instead, embryonic NSC-derived cell populations expressed Shh and were indistinguishable from endogenous regenerated tail ET cells under control conditions, suggesting the loss of roof plate and ventralization of embryonic NSC identities (Fig. 6B, D). Reconstitution of regenerated tail ETs by exogenous embryonic and adult NSCs was not affected by cyclopamine (Fig. 6B–D). However, cyclopamine

treatment did affect the abilities of ET regions derived from exogenous embryonic NSCs to retain roof plate identity and differentiate into neurons (Fig. 6B, D). Tails regenerated by lizards implanted with DiI-labeled exogenous embryonic NSCs and treated with cyclopamine exhibited characteristic Fig. 8 double ETs consisting of dorsal DiI-labeled exogenous tubes and ventral unlabeled endogenous tubes (Fig. 6B), perhaps indicating a resistance of embryonic NSCs to fully integrate with adult NSC structures when protected from endogenous Hedgehog signaling. Only DiI-labeled ET regions expressed Pax7 and Tuj1 and not Shh (Fig. 6B, D). Instead, Shh expression was limited to endogenous DiI$^-$ ET regions, resulting in patterned lizard ETs exhibiting dorsal Pax7$^+$ Shh$^-$ roof plates and ventral Pax7$^-$ Shh$^+$ floor plates (Fig. 6B). Patterned ETs were only observed in conditions concerning both embryonic NSCs and cyclopamine treatment (Fig. 6B). Otherwise, all other exogenous cell progenies fully incorporated into the endogenous ET cell populations. In particular, exogenous adult NSCs contributed to ET floor plate identities only and were unresponsive to native Hedgehog signaling environments (Fig. 6C, D). Taken together, these experiments created regenerated lizard tails with dorsoventrally patterned ETs and newly differentiated neurons. Relief from endogenous Hedgehog signaling allowed embryonic NSC populations to recreate NT-like regions within the regenerated lizard tail ET.

**Embryonic NSCs resistant to Hedgehog pattern regenerated tails**. The goal of this study was to induce dorsoventral patterning in both CNS and skeletal tissues, but since Hedgehog signaling was also responsible for induction of blastema cell cartilage differentiation, systemic cyclopamine treatments used in above experiments were unsatisfactory because they also inhibited skeletogenesis. We hypothesized that a more targeted approach was needed; embryonic NSCs genetically altered to be unresponsive to Hedgehog signaling would retain roof plate identity and resist ventralization following incorporation into the Shh-saturated regenerated lizard tail environment. The simultaneous repression of Shh signaling and enhancement of NSC roof plate differentiation capacity would induce dorsoventral patterning in the lizard ET and, hence, regenerated cartilage (Fig. 7).

Intracellular Hedgehog signaling passes through the essential membrane protein Smo, often referred to as the switch of Hedgehog signaling[18]. Indeed, affecting Smo activity is the most efficient means of modulating Hedgehog signaling, and both cyclopamine and SAG target Smo to induce inhibition and activation, respectively[19,20]. Without Smo, cells become unresponsive to Hedgehog signaling even when stimulated by high levels of Hedgehog ligands. For these reasons, we chose to create embryonic NSCs that are unresponsive to Hedgehog signaling by knocking down Smo. Here we used CRISPR/Cas9 gene editing to replace *Smo* with *GFP* genes in embryonic NSCs. First, the *Lepidodactylus lugubris Smo* gene (*LlSmo*) was sequenced and used to design gRNA to exon 2 (*LlSmo*-gRNA) (see Supplementary Table 1 for sequences). *LlSmo* gRNAs were tested for their efficacy in knocking out *LlSmo* and rendering embryonic NSCs unresponsive to Hedgehog stimulation in vitro and in vivo. Homology-directed repair (HDR) donor template plasmids contained appropriate left and right homology arm sequences specific for each gRNA-directed double-strand breaks and included GFP gene inserts. Promoter-free GFP HDR inserts were used to limit background fluorescence produced by unintegrated HDR plasmids, thereby eliminating the need for episomal dilution through repeated passaging prior to sorting. Single-cell suspensions of embryonic lizard tail NSCs were transfected with Cas9/*LlSmo*-gRNA/HDR complexes, and FACS was used to

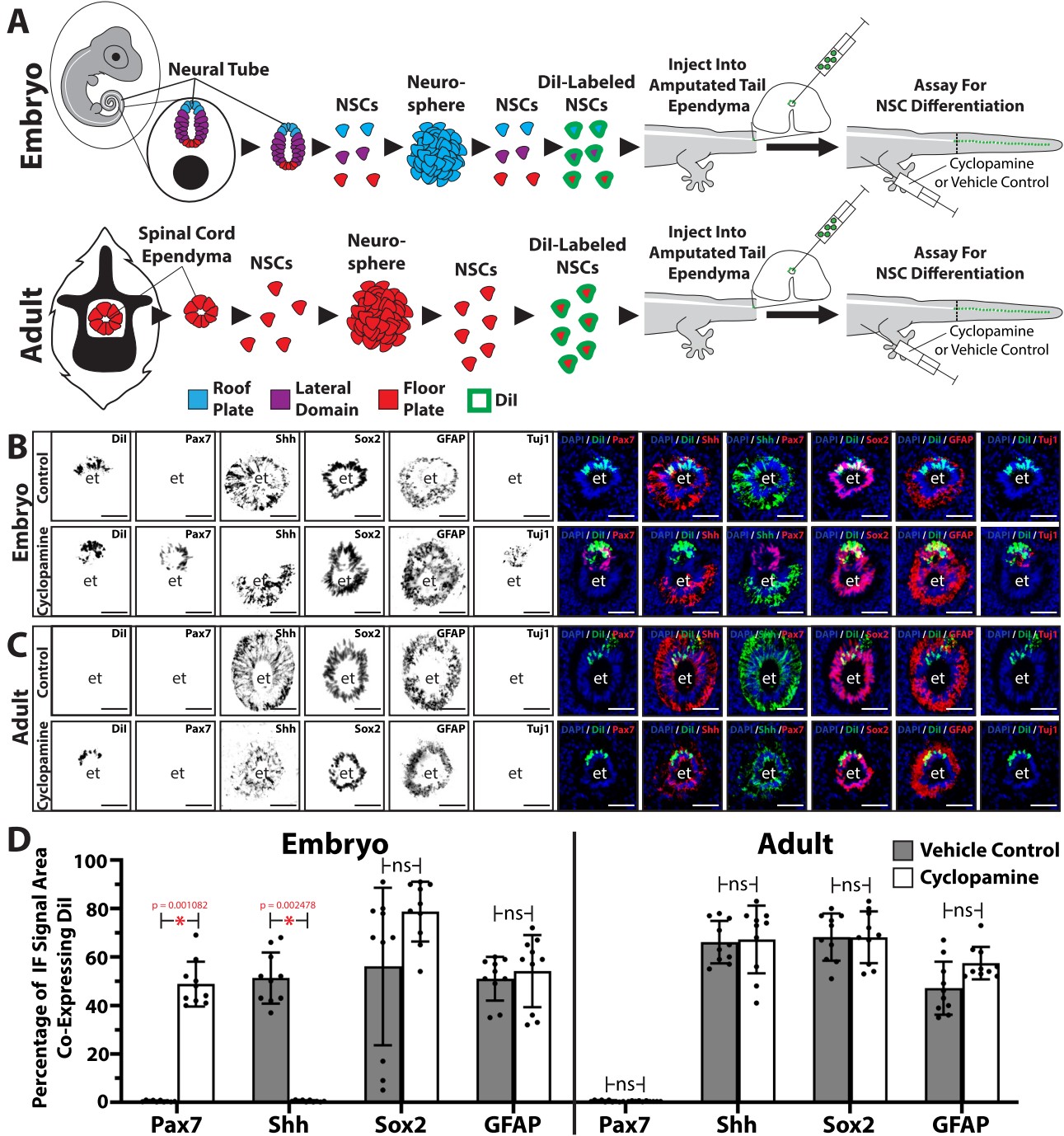

**Fig. 6 Embryonic lizard tail NSCs introduced into adult tails contribute to the regenerated ETs and are ventralized by the native Hedgehog signaling environment. A** Experimental scheme for introducing embryonic and adult NSCs into regenerated tail ETs. NSCs were isolated from embryo NTs and adult spinal cord ependyma, cultured as neurospheres, dissociated and labeled with DiI. DiI-labeled cells were injected into dorsal regions of adult spinal cord ependyma of tails that had been amputated. Lizards were treated with cyclopamine or vehicle control for 4 weeks while tails regenerated. Regrown tails were collected, and the differentiation and patterning of DiI-labeled exogenous cells were analyzed. **B**, **C** Cross sections of ETs of tails grown from stumps implanted with **B** embryonic or **C** adult NSCs. For each type of NSC, images of tails treated with vehicle control are depicted in rows on top of cyclopamine-treated tail images. For all rows, individual signal channels are presented to the left, and merged images of select channels are presented on the right. et ependymal tube. Bar = 100 μm. **D** Quantification of exogenous embryonic and adult NSC differentiation in response to treatment with cyclopamine in vivo. Embryonic and adult NSCs pre-labeled with DiI were injected into dorsal regions of tail spinal cord ependyma. Following amputation and treatment with vehicle control or cyclopamine for 28 days, regenerated tails were collected and processed by histology/IF/florescence microscopy every 1 mm along entire tail lengths. Percentages of DiI+ ET areas co-expressing Pax7, Shh, Sox2, or GFAP are presented. n = 10 different animals/tails for each condition. Data are presented as mean values +/− SD. Two-way ANOVA with pairwise Tukey's multiple comparison tests was used. *p < 0.01 compared to corresponding vehicle control conditions; ns, not significant. Source data are provided as a Source data file.

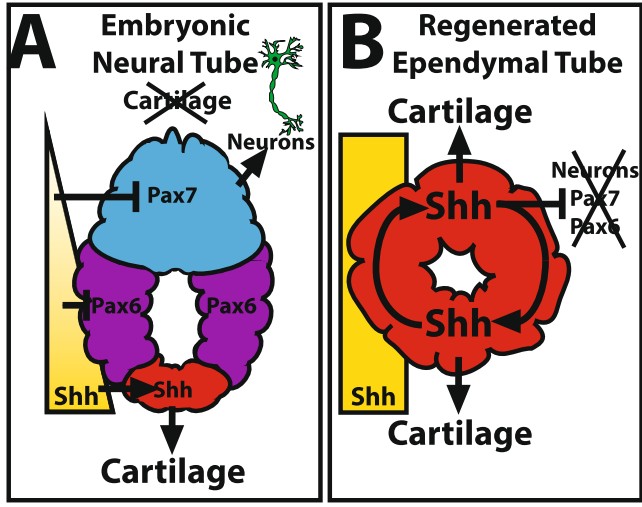

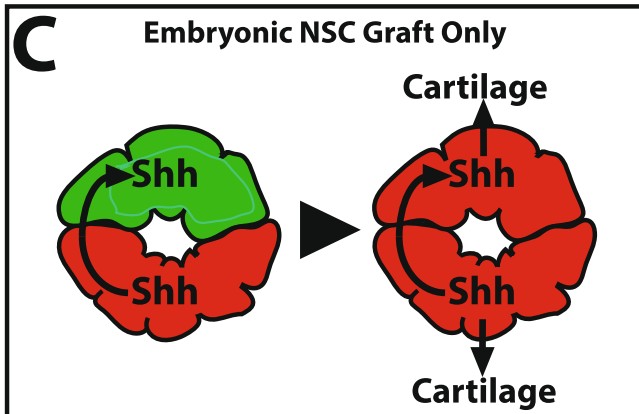

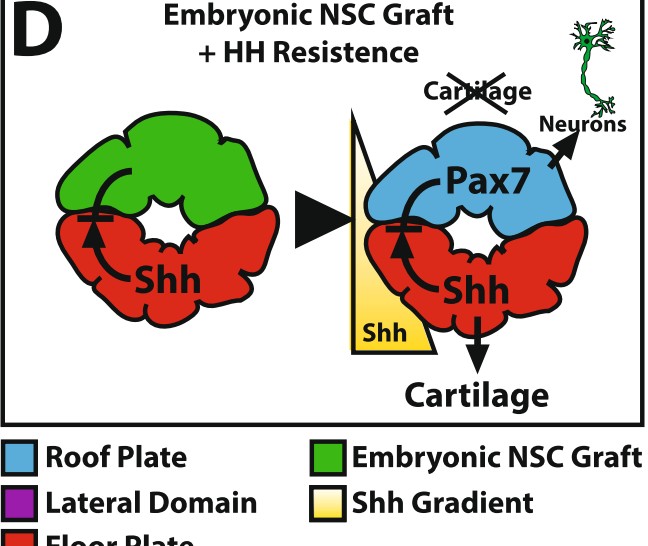

**Fig. 7 Schematic representation of the dorsoventral patterning signals present during lizard tail development and regeneration. A** Embryonic lizard tail neural tubes contain Pax7[+] roof plate, Pax6[+] lateral domains, and Shh[+] floor plates. Hedgehog signaling induces embryonic NSC ventralization and neurogenesis in addition to cartilage formation in surrounding cells. Shh expression restricted to floor plates results in Hedgehog signaling gradients that maintain dorsoventral patterning of cartilage formation and neural tube differentiation. **B** Adult regenerated tail ependymal tubes lack roof plates and lateral domains are entirely made up of floor plate. Absence of Shh gradients results in unregulated Shh expression that abolishes cartilage dorsoventral patterning and inhibits Pax7 and Pax6 expression and neurogenesis. **C** Embryonic NSC grafts are ventralized by endogenous Shh signals produced by ependymal tubes, resulting in unpatterned NSC populations and chondrogenesis. **D** To create dorsoventrally patterned regenerated lizard tails, embryonic NSCs pre-engineered to be resistant to hedgehog signaling were engrafted to dorsal ependymal tube regions. Protected from endogenous Shh signals, exogenous NSCs maintained roof plate identities, differentiated into neurons, and induced dorsoventral cartilage patterning.

RT-PCR (Supplementary Fig. 13) using primers to *LlSmo* (see Supplementary Table 1 for sequences). All 50 *Smo*-CRISPR clonal lineages exhibited significantly lower *Smo* mRNA levels than those expressed by Control-CRISPR NSCs (Supplementary Fig. 16). *Smo*-CRIPSR NSC populations exhibited a range of *Smo* mRNA amounts, and some lineages expressed undetectable levels. *Smo*- and Control-CRIPSR clonal populations were also stimulated with SAG and analyzed for Gli1 mRNA expression via real-time RT-PCR (Supplementary Fig. 17) and Gli1 promoter activity via luciferase-based Gli1/Hedgehog pathway reporter assays (Supplementary Fig. 18). SAG treatment induced significant upregulation in Gli1 mRNA expression and Gli1 promoter activity in Control-CRISPR cells (Supplementary Figs. 17, 18). Among *Smo*-CRISPR cells, several populations exhibited unresponsiveness to SAG stimulation (Supplementary Figs. 17, 18). These Hedgehog-resistant clonal populations were the same ones that exhibited the lowest *Smo* mRNA expression levels (Supplementary Fig. 16). Taken together, these in vitro assays validated and identified *Smo*-CRISPR embryonic NSC populations in which *Smo* genes were replaced by GFP and Hedgehog signaling was interrupted.

The clonal *Smo*-CRISPR embryonic NSC population with the lowest *Smo* expression and weakest responses to exogenous Hedgehog signals was chosen for in vivo experiments (Fig. 8). These *Smo* knockout (KO) NSCs were tested for their differentiation potential and effects of tissue dorsoventral patterning during lizard tail regeneration (Fig. 8A). *Smo* KO neurospheres were trypsonized to yield single-cell suspensions and injected into the dorsal regions of amputated tail ependyma (Fig. 8A). Control tails received injections of DiI-labeled embryonic NSCs derived from Control-CRISPR neurospheres. After 4 weeks regeneration, tail samples were collected and analyzed for GFP/DiI fluorescence to trace contributions of injected cells to regenerated tissues and for expression of markers Col2, Tuj1, Sox2, Shh, and Pax7 to determine effects on regenerated tissue differentiation and patterning (Fig. 8D–Z). Tails collected from lizards treated with either *Smo* KO or control NSCs achieved similar lengths (1.2 cm) and exhibited ETs with fluorescently labeled dorsal regions (Fig. 8D–X) (Supplementary Fig. 19), indicating that *Smo* KO did not interfere with exogenous NSC engraftment and contribution to regenerated ependyma populations. As observed with embryonic NSC transplants during cyclopamine treatment, *Smo* KO embryonic NSCs contributed to the dorsal halves of distinct figure-eight double ETs (Fig. 8D, H,

**Legend:**
- Roof Plate (blue)
- Lateral Domain (purple)
- Floor Plate (red)
- Embryonic NSC Graft (green)
- Shh Gradient (yellow)

isolate populations of GFP[+] cells representing successfully edited *Smo*-CRISPR NSCs (Fig. 8A) (see Supplementary Fig. 15 for FACS gating strategy). This method yielded an editing efficiency of 90% (Fig. 8B). Control-CRIPSR NSCs consisted of identically treated cell populations that were transfected with nonbinding scramble gRNA rather than *LlSmo*-gRNA and served as controls for *Smo*-CRISPR NSCs. *Smo*-CRISPR cells derived from individual GFP[+] neurospheres (Fig. 8C) were expanded and passaged to yield 50 near-clonal lineages. *Smo*-CRISPR clonal populations were individually tested for *Smo* RNA expression by real-time

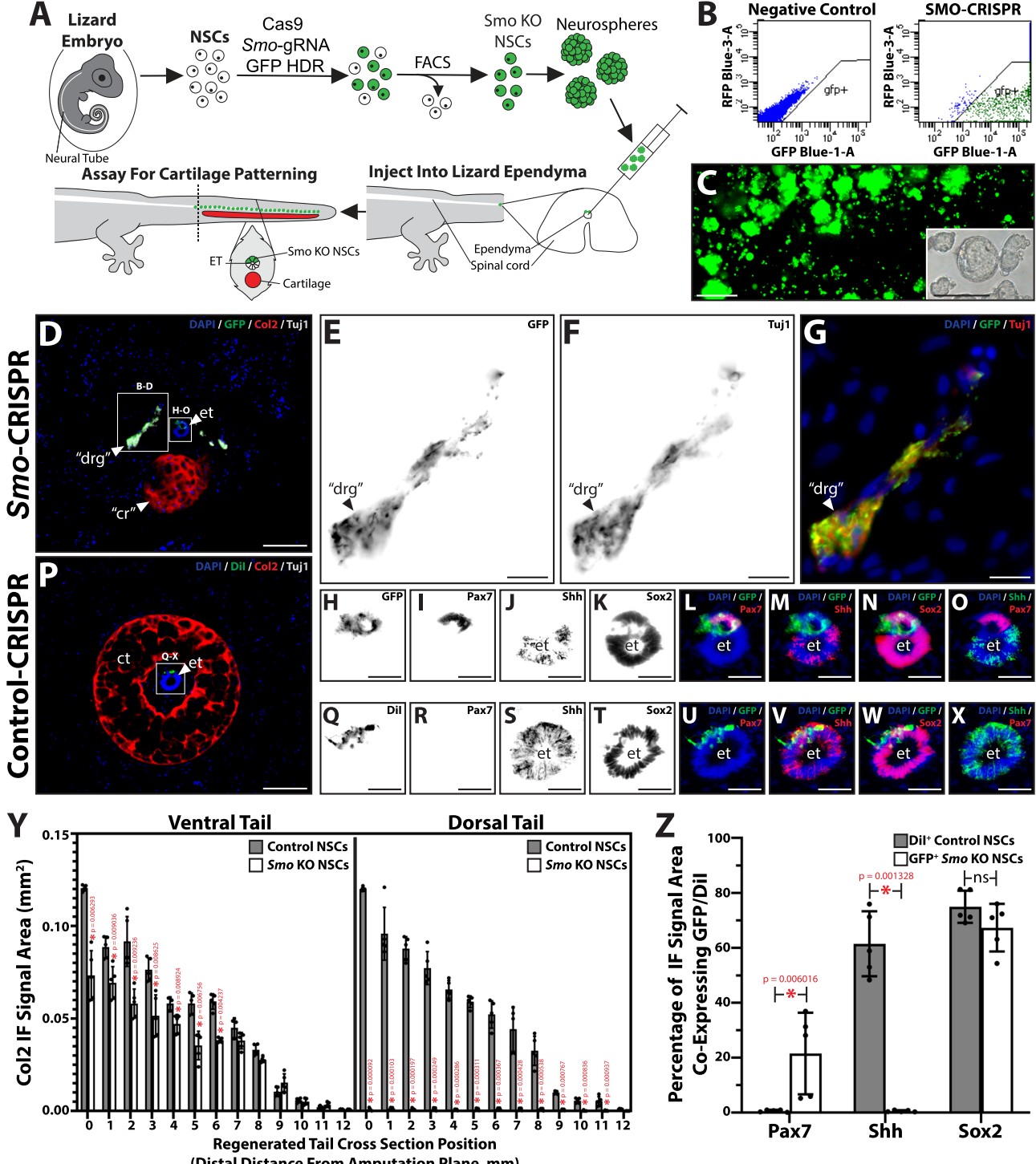

L–O). Interestingly, tails that received injections of *Smo* KO NSCs exhibited Col2+ cartilage rods ventral to ETs (Fig. 8D) rather than the cartilage tubes that completely surround ETs in control tails (Fig. 8P). Quantification of dorsal and ventral Col2 immunofluorescence (IF) signals along tail lengths indicated that the presence of dorsal ET regions derived from exogenous *Smo* KO embryonic NSCs significantly reduced dorsal and ventral cartilage areas compared to control tails (Fig. 8Y). Dorsal regions of tails regrown by *Smo* KO NSC-treated lizards also exhibited Tuj1+ DRG-like neural structures (Fig. 8D–G) that were absent from control tails (Fig. 8P) (Supplementary Fig. 20). Unlike cartilage regions, ectopic DRG-like structures co-expressed GFP,

indicating their derivation from exogenous *Smo* KO embryonic NSC implants (Fig. 8D–G). Analysis of patterning markers revealed that ETs regenerated in animals treated with *Smo* KO NSCs exhibited dorsoventral patterning (Fig. 8H–O), while those of control tails did not (Fig. 8Q–X). Dorsal regions derived from *Smo* KO, but not control, NSCs expressed Pax7 (Fig. 8I, R, Z). Pax7 expression co-localized with dorsal GFP+ regions (Fig. 8L, Z), while Shh expression was excluded and restricted to GFP− ventral endogenous ET regions (Fig. 8M, Z). The resultant ETs regenerated in *Smo* KO NSC-treated lizard bore strikingly resemblances to embryonic ETs, with GFP+ Shh− Pax7+ dorsal roof plates and GFP− Shh+ Pax7− ventral floor plates

**Fig. 8 Gene-edited *Smo* KO neural tube NSCs recapitulate embryonic differentiation and dorsoventral patterning in adult regenerating lizard tails.**
**A** Experimental scheme for introducing *Smo* KO embryonic NSC populations into regenerating lizard tail ETs toward inducing dorsoventral patterning of regenerated tissues. NSCs were isolated from embryonic tail NTs and edited to replace *Smo* gene with GFP by transfecting with Cas9 protein, *Smo*-gRNA, and GFP HDR template. *Smo* KO NSCs were enriched by FACS, propagated as neurospheres, and injected into dorsal regions of spinal cord ependyma in amputated adult lizard tails. *Smo* KO NSC populations engrafted and contributed to dorsal regions of regenerated tail ETs, and regrown tails were assayed for effects on cartilage patterning. **B** FACS analysis of embryonic NSCs treated with Cas9, *Smo*-gRNA, and GFP HDR. Editing efficiency was ~90%. **C** GFP fluorescence micrograph of clonal *Smo* KO NSC neurospheres. **C** (Inset) Brightfield micrograph of *Smo* KO neurospheres. **D** Representative cross section of a tail regenerated from a lizard implanted with embryonic *Smo* KO NSCs and analyzed for GFP, Col2, and Tuj1 expression. **E–G** Higher magnification view of region identified in Panel D containing ectopic neural structures analyzed by GFP and Tuj1 IF. **H–O** Higher magnification views of ET region identified in Panel D analyzed by GFP, Pax7, Shh, and Sox2 IF. **P** Representative cross section of a regenerated tail produced by a lizard implanted with control NSCs treated with scramble gRNA and pre-labeled with DiI. **Q–X** Higher magnification views of ET regions identified in Panel P analyzed by GFP, Pax7, Shh, and Sox2 IF. **Y** Quantification of dorsal and ventral Col2 IF signal areas along regenerated tail lengths following treatment with *Smo* KO or control NSCs. $n = 5$ distinct animals/tails per condition. Data are presented as mean values $+/-$ SD. **Z** Quantification of Pax7, Shh, and Sox2 expression in GFP$^+$ *Smo* KO and DiI$^+$ control NSC following incorporation into adult regenerated tails. $n = 5$ animals/tails per condition. Data are presented as mean values $+/-$ SD. Two-way ANOVA with pairwise Tukey's multiple comparison tests was used. $*p < 0.01$ compared to corresponding control conditions. Source data are provided as a Source data file. ns not significant, cr cartilage rod, ct cartilage tube, drg dorsal root ganglion, et ependymal tube. Bar $= 50\,\mu m$ (**D** and **P**). Bar $= 10\,\mu m$ (**C**, **E–O** and **Q–X**).

(Fig. 8L–O). ETs formed in lizards treated with Di-labeled control NSCs expressed Shh throughout, including dorsal DiI$^+$ regions, and did not form roof plate domains (Fig. 8U–X). These results indicated that ET cells derived from *Smo* KO NSCs differentiated into roof plate structures and neurons and introduced dorsoventral patterning into regenerated CNS and skeletal tissues.

## Discussion

Lizards evolved more than 150 million years ago, and yet no lizard species has ever developed the ability to regenerate tails with dorsoventrally patterned tissues[21]. Using a combination of embryonic stem cell transplantations, CRISPR/Cas9 genome editing technology, and a unique parthenogenetic lizard species, we created regenerated lizard tails with dorsoventrally patterned CNS and skeletal tissues. In doing so, we uncovered the effects of NSC identities and differentiation capacities on tail regenerative outcomes. Regenerative programs are often presented as recapitulating developmental processes to replace lost tissues. For example, during embryonic tail development, NTs differentiate into CNS tissues and provide Shh patterning signals that induce skeletogenesis in tail bud mesoderm. Similarly, during tail regeneration, ETs reform CNS tissues and direct skeletal differentiation in blastema cells via Hedgehog signaling. In some animals, like salamanders and frog tadpoles, CNS differentiation and skeletal patterning during tail regeneration faithfully recapitulates embryonic tail developmental processes. Lizard tails, on the other hand, present a unique case in which regenerative outcomes fail to recreate embryonic tail formation. This study suggests that lizard tail regeneration is unable to recapitulate the CNS differentiation and skeletal patterning achieved during embryonic tail development due to limited differentiation capacity of adult NSCs that prevents roof plate differentiation. Embryonic NSCs are able to differentiate into Pax7$^+$ Shh$^-$ roof plate domains in tail NTs that effectively limit Shh expression and subsequent Hedgehog signaling to floor plates and ventral tail regions. Roof plate NSCs undergo neurogenesis and give rise to DRGs, while Shh produced by floor plate induce chondrogenesis/skeletal development in ventral tail bud cells. In contrast, adult lizard NSCs are unable to form roof plates, resulting in uniform floor plate identity, unregulated Shh production, and, ultimately, unpatterned cartilage formation during tail regeneration.

We hypothesized the key to introducing dorsoventral patterning in the adult regenerated lizard tail is to recreate the Shh$^-$ roof plate/Shh$^+$ floor plate spatial expression profile characteristic of the embryonic tail NT. This was accomplished by directly transplanting embryonic NSCs derived from tail NTs into adult tail spinal cords, where they contributed to dorsal regions of regenerated tail ETs. These experiments were only possible through the use of the parthenogenetic lizard species *L. lugubris*, one of the only such lizards that is both diploid (which facilitates genome editing protocols) and capable of regenerating its tail, and one of the few examples of obligate parthenogenesis among vertebrates[22]. This all-female species of gecko, which reproduces asexually to generate genetically identical clonal offspring, exhibits no detectable intra-clonal variation in MHC genes[23], and cells and tissues can be transplanted among animals within clonal lineages without rejection[14,22,24]. The clonal characteristics of *L. lugubris* reproduction means that embryonic NSCs and their progeny are able to be implanted into any adult animal without rejection and without the use of immunosuppressants, which have been shown to adversely affect developmental and wound healing processes, including regeneration[25–27].

Embryonic NSCs and their progeny were capable of roof plate differentiation when implanted into adult tails, but unchecked Hedgehog signaling effectively ventralized any NSC population introduced to the lizard ET. We hypothesized that only the simultaneous repression of Shh signaling and enhancement of NSC roof plate differentiation capacity would induce patterning in the lizard ET and, hence, regenerated cartilage. *Smo* KO embryonic NSCs generated via CRISPR/Cas9 gene editing were unresponsive to Hedgehog stimulation and retained roof plate identity and potential for neurogenesis. When incorporated into dorsal regions of adult regenerated tail ETs, *Smo* KO NSCs resisted ventralization, differentiated into roof plate structures and neurons, and prevented floor plate Shh signals from inducing chondrogenesis in dorsal tail regions. These edited embryonic NSCs were sufficient for recreating developmental dorsoventral patterning in adult regenerated tails. Future work will focus on the functional ramifications of neurogenesis and skeletal patterning introduced to regenerated lizard tails.

This study outlines innovative workflows to genetically manipulate tail NSC populations through the introduction of exogenous cells gene-edited in vitro. In other, more traditional model organisms, this work would typically be accomplished through the creation of stable transgenic lines. However, aspects of reptile reproductive biology present hurdles for generating stable transgenic lizard lines. Reptiles are generally slow to reach sexual maturity and/or reproduce seasonally in response to environmental triggers. Many species are also difficult to maintain in captivity and lose fecundity without revitalization from outbreeding with wild-caught stock. Finally, reptilian oviposition takes place at advanced stages of embryogenesis compared to

birds, precluding the application of transgenesis protocols established for chickens to lizards. Despite these hurdles, the first transgenic lizard line was recently produced through direct injection of Cas9 ribonucleoprotein complexes into ovarian follicles of the brown anole lizard (*Anolis sagrei*)[28], a sexually-reproducing species. This tremendous achievement certainly provides proof-of-concept for developing transgenic lizard models for investigating a wide range of biological processes in lizards. However, the realities of investigating adult lizard biological processes, such as lizard tail regeneration, are not compatible with the current state-of-the art of lizard transgenesis. For example, gene knockdowns are necessary to investigate the specific molecular regulators of tail regeneration. Lizard tail regeneration involves many of the same signaling cascades involved in embryonic tail development, including Hedgehog signaling. Constitutive organismal knockouts of members of these fundamental pathways, such as Smo, would not survive embryonic development, limiting their usefulness in investigating adult wound healing processes. Unfortunately, the inducible knockouts that are needed to avoid these complications are beyond current lizard transgenesis capabilities, and the realities of lizard sexual reproduction means time commitments of multiple years for the generation of each of the required lines. Thus, there was a need to directly target regenerating lizard tail cells with genetic modifications and manipulations. Previous studies have applied transfection[29,30] and transduction[31] methods to introduce exogenous sequences coding for mRNA or shRNA to the already-formed blastema. However, these techniques were non-specific, making it impossible to target specific cell populations, and impermanent, suffering from episomal dilution in highly proliferative cell populations. Here we overcame these shortcomings and described a platform for introducing gene-edited NSCs to specific regions of regenerated tail ETs in the asexually reproducing lizard *L. lugubris*.

This study also introduces viral vector-based methods for conducting lineage traces of embryonic lizard cell populations through development and regeneration. In doing so, we highlighted interesting aspects of DRG formation unique to the lizard tail, the only amniote tail that contains both spinal cord and DRG. For example, in the trunks of amniote embryos, folding of Sox2+ neural plates gives rise to lateral and floor domains of primary neural tubes, while neural plate borders pinch off and contribute to neural tube roof plate domains and Sox2− neural crest[32–34]. Trunk neural crest cells migrate mediolaterally and begin to express Sox2 before forming DRG[33,34]. The situation is very different in amniote tails, and even more exceptional in lizard tails. Tails are formed from tail buds, which do not have neural plates or neural folds, and hence, no neural crest cells[32]. Instead, Sox2+ neural mesodermal progenitors (NMPs) derived from Sox− tail bud cells directly differentiate into secondary neural tubes[32]. Mammalian secondary neural tubes degenerate before DRG formation during tail development, and adult mammal tails contain neither DRG nor spinal cord[35]. In contrast, lizard secondary neural tubes survive tail development, and lizards are the only amniotes that exhibit both spinal cords and DRG in adult tails. The origins of amniote tail DRG has not been specifically studied, but lineage tracing data presented here suggest that lizard tail DRG derive from Sox2+ tail bud cells that pass through a neural tube intermediate. Future work will focus on determining mechanisms underlying divergent tail CNS development and persistence of NMP-derived tail cell populations between lizards and other amniotes.

In summary, regenerated lizard tails with dorsoventrally patterned spinal cord and skeletal tissues have been created. In reaching this milestone, this study provides insight into the principles governing the fidelity with which regenerative processes recapitulate embryonic development. Specifically, the ramifications of NSC cell states at times of initial tail amputations on final regenerative outcomes are highlighted. In salamanders, embryonic tail NT dorsoventral patterning is maintained into adulthood, and adult salamander spinal cord NSC populations exhibit roof plate and floor plate identities that carry over to ETs during tail regeneration[14,36]. Conversely, lizard tail NT dorsoventral patterning is not maintained during embryonic development, and adult lizard tail ependyma exhibit floor plate identity only. Upon adult tail amputation, these lineage-restricted populations are the only NSCs available to reconstitute regenerated tail ETs, resulting in the loss of dorsoventral patterning in regenerated tail tissues. This study demonstrated that regenerated lizard tail patterning can be restored if endogenous adult spinal cord ependyma cell populations are supplemented with exogenous, roof-plate competent NSC populations at the onset of tail regrowth. While anamniotic salamanders and frogs are able to turn back the clock and reform embryonic structures in adult ectoderm-derived tissues[14,36,37], amniotic lizards have lost this ability. This sacrifice in cell plasticity is one of the trade-offs between evolution and regenerative potential[11,38], but this study highlights ways that this trend can be reversed. Future work will investigate the epigenetic changes that occur in lizard tail NSC populations through development and into regeneration with the aim of identifying the specific genetic regions responsible for divergent roof plate differentiation capacities.

## Methods

All reagents/chemicals were purchased from Sigma-Aldrich unless otherwise specified. Additional materials and methods can be found in Supplementary Information. All experiments complied with relevant ethical regulations for animal testing and research.

**Lizards.** All experiments were carried out with the mourning gecko *Lepidodactylus lugubris*. This parthenogenetic, all-female lizard species reproduces asexually to yield clonally identical offspring[22]. Cells and tissues can be transplanted among colony members without the use of immunosuppressant and anti-rejection therapeutics[14], which have been shown to negatively affect tail regeneration. *L. lugubris* is believed to be a hybrid species, and more than 20 clonal populations are spread throughout islands of the Pacific and Indian Oceans, several of which exhibit polyploidy[22]. The founding members of our research colony were collected from Hawaii, which we verified as belonging to a diploid clonal population via karyotyping (service provided by the Molecular Cytogenetics Laboratory, Department of Veterinary Integrative Biosciences, Texas A&M University) (Supplementary Fig. 18). All lizard experiments were performed according to the guidelines of the Institutional Animal Care and Use Committee at the University of Pittsburgh (protocols 15114947, 16128889, and 18011476) and the University of Southern California (protocol 20992).

**Tail sample collection.** Embryonic lizard tails were collected 7 and 14 days post-oviposition (DPO). Adult lizard tails were collected 28–112 days post hatching (DPH). Regenerated tail samples were collected 28 days post-tail amputation (DPA). Ethyl chloride spray was used to numb surgical sites, and tail amputations were performed with number 22 scalpel blades.

**Immunostaining.** Lizard and salamander tissue samples were analyzed by IF as previously described[13]. The following primary antibodies were used: Col2 (Abcam ab34712, 1:1000); Sox2 (Abcam ab97959, 1:1000); Tuj1 (Abcam ab78078, 1:1000); Shh (Novus Biologicals NBP1-69270, 1:1000); Pax7 (Developmental Studies Hybridoma Bank PAX7-c, 1:500; BMP4 (Abcam ab118867, 1:1000); Pax6 (Developmental Studies Hybridoma Bank PAX6-s, 1:200); FoxA2 (Developmental Studies Hybridoma Bank 4C7, 1:100); GFAP (Abcam ab4674, 1:1000); NeuN (Abcam ab177487, 1:1000). All IF images of sagittal sections are presented dorsal toward the top, ventral toward the bottom, distal toward the right, and proximal toward the left. Transverse sections are presented with dorsal on top and ventral on bottom. Positive IF staining areas were quantified using Fiji (Image J, NIH) applying over/under thresholding to limit analysis to positive staining areas.

**Drug treatments.** Embryonic and adult lizards were treated with the Hedgehog inhibitor cyclopamine or the smoothened (Smo) agonist SAG in vivo. To treat lizard embryos, a microinjector system was used to deliver drug solutions in ovo (Supplementary Fig. 6). Lizard eggs were injected with 3 μl of 500 μg/ml cyclopamine or 800 μg/ml SAG. To treat adult lizards, cyclopamine (50 μg/g) or SAG

(40 μg/g) were administered via intraperitoneal (i.p.) injections every 24 h. In vitro, neurosphere cultures were treated with 10 μM cyclopamine or 500 nM SAG.

**Isolation of lizard NSC populations and generation of neurospheres**. Tails were collected from lizard embryos (14 DPO), adult lizards with original tails (28–112 DPH), and from adult lizards with regenerated tails (28 DPA). NTs, spinal cords, and ETs were microdissected from embryonic, adult, and regenerated tails, respectively. NTs and ETs were digested in L15 medium (Gibco) containing 30 U/mL papain, 0.5 mg/mL BSA, 0.24 mg/mL cysteine, 40 μg/mL DNase I type IV, and 1.0 mg/mL trypsin inhibitor for 30 min at room temperature, while adult spinal cords were digested for 1 h. Digested tissues were homogenized by repeatedly passing solutions gently through 1 mL pipette tips. Digestion was halted with ovomucoid inhibitor [1.0 mg trypsin inhibitor, 0.5 mg/mL BSA, and 40 mg/mL DNase I type IV in L15 medium (Gibco)]. Cell suspensions were added to neurosphere culture medium [2 μg/mL heparin, 20 ng/mL bFGF (RayBiotech, Norcross, GA, USA), 1x ITS (Gibco), and 1x B-27 (Gibco) in DMEM/F12 + Glutamax medium (Gibco)] with 1x Pen/Strep added (Gibco), filtered through 100 μm filters, and centrifuged for 5 min at $380 \times g$. Cell pellets were resuspended in 0.9 M sucrose solutions and centrifuged at $750 \times g$ for 30 min. Myelin was carefully aspirated, and pellets were resuspended in neurosphere medium and plated at densities of ~40,000 cells/well. Dorsoventral marker analysis and drug treatment without differentiation were performed with freshly isolated neurospheres (passage 0), while in vitro differentiation assay, in vivo implantation experiments, and CRIPSR/Cas9 protocols were performed with passage 3 neurospheres. To passage neurospheres, cultures were incubated with 0.25% Trypsin at 37 °C for 3 min. 2 mM CaCl₂ and 40 μg/mL DNase I type IV were added, and neurospheres were dissociated by gently pipetting up and down with 200 μl tips. To end dissociation, 4 mg/ml trypsin inhibitor was added, and neurospheres were washed in neurosphere media before re-plating at 100,000 cells per well.

**Tracing lineage contributions of Sox2⁺ NT cells through tail development and regeneration**. A combination of lentiviral and adeno-associated virus (AAV) vectors was used to deliver CreStoplight and Sox2 promoter-driven Cre recombinase (Sox2-Cre) constructs, respectively, to embryonic lizard tails. Lentiviral Sox2-Cre constructs consisting of Cre genes downstream of 1 kb inserts of the lizard (*L. lugubris*) Sox2 promoter region were packaged into AAV6 viral particles (see Supplementary Information Table 1 for *L. lugubris* Sox2 promoter insert sequence). CreStoplight constructs consisted of adjacent green fluorescent protein (GFP) and transcription terminator (Stop) sequences under transcriptional regulation of constitutively active CAG promoters. GFP and Stop elements were flanked by loxP and loxH sites and followed by red fluorescent protein (RFP) inserts[39]. CreStoplight constructs were packaged in VSV-G pseudotyped lentiviral envelopes. Under this Sox2-Cre/CreStoplight system, transduced cells were labeled with GFP in the absence of Cre but were irreversibly converted to RFP labeling upon the co-expression of Cre in Sox2⁺ embryonic NSCs. Two different viral vectors were used to modulate the persistence of each packaged insert in lineages derived from transduced cells. Lentivirus was chosen to deliver CreStoplight constructs because it would provide permanent genomic integration of GFP/RFP genes, allowing the fates of embryonic Sox2⁺ cell progenies to be traced through development and regeneration. Conversely, AAV6 was chosen to deliver Sox2-Cre constructs as episomes that would be diluted through repeated cell divisions, allowing for targeted Cre-labeling of embryonic and not adult Sox2⁺ NSC populations. The abilities of this dual vector system to provide transient Cre expression and prolonged GFP/RFP labeling were simultaneously validated via Western blots (Supplementary Fig. 5).

To perform lineage tracing, viral vectors were injected into the amniotic sacs of freshly laid lizard eggs (0 DPO) using a microinjection system (Supplementary Fig. 6). Embryos were collected at 7 and 14 DPO each and analyzed for RFP, GFP, Sox2, and TUJ1 expression (Supplementary Figs. 7, 8, 11) as well as Pax7, Pax6, and Shh expression (Supplementary Fig. 12). To determine lineage contributions to adult and regenerated tails, embryos transduced with the Sox2-Cre/CreStoplight hatched and matured for 28 days post hatching (DPH) before their tails were collected by amputation. The resultant regenerated tails were then re-collected 28 days post-amputation (28 DPA). Samples of both original and regenerated tails were analyzed for RFP, Sox2, Tuj1, GFAP, Pax7, Pax6, and Shh expression (Supplementary Figs. 10–12).

**RNA isolation and real-time RT-PCR**. Total RNA of cultured lizard neurospheres was isolated and purified using RNeasy Plus Mini Kits. Reverse transcription reactions were performed using Superscript® VILO™ cDNA Synthesis Kit (Invitrogen) according to the manufacturer's manual. Real-time PCR was performed using the SYBR Green Reaction Mix (Applied Biosystems, Foster City, CA, USA) with a StepOne-Plus thermocycler (Applied Biosystems). See Supplementary Table 1 for real-time RT-PCR primer sequences. All sample values were normalized to GAPDH using the 2-ΔΔCt method.

**CRISPR/Cas9 gene editing**. Neurospheres derived from embryonic tail NTs were passaged to yield single-cell suspensions. The Alt-R CRISPR-Cas9 system (Integrated DNA Technologies) was used to deliver Cas9/gRNA/HDR template

complexes into embryonic NSCs using the 4D-Nucleofector platform with 96-well shuttle attachment (Lonza) according to the manufacturers' instructions. Briefly, *LlSmo*-gRNA complex was formed by combining *LlSmo* crRNA and tracrRNA, incubated at 95 °C for 5 min, and annealing at room temperature for 20 min. Cas9:*LlSmo*-gRNA ribonuclear (RNP) complex was formed by incubating 4 μM *LlSmo*-gRNA complex with 4.8 μM Cas9 solution for 20 min at room temperature. Transfection reaction solutions were prepared by combining 20 μl of embryonic NSC suspension with 5 μl RNP complex, 1.2 μl of HDR donor plasmid (100 μM), 1.2 μl electroporation enhancer solution (100 μM) (IDT), and 2.6 μl PBS and transferred to 96-well Nucleocuvette modules (Lonza). Nucleofector program T-023 was used to electroporate cells. After transfection, cells were added to pre-warmed neurosphere media containing 30 μM HDR enhancer reagent (IDT). After 14 days of culture, neurospheres were dissociated, and GFP⁺ NSC populations were selected via FACS with a BD FACS Aria II flow cytometer. Selected single cells were pooled and cultured for 2 weeks to neurospheres. Fifty individual neurospheres were transferred to 96-well plates and passaged separately as the clonal lineages referenced in this study.

**Hedgehog activity luciferase assays**. NSC neurospheres were co-transfected with pMuLE_ENTR_12GLI-FLuc_R4-R3 and RL-CMV plasmids (Addgene) using a 4D-Nucleufector platform with 96-well shuttle attachment (Lonza). Following 24 h of recovery, cells were treated with 100 nM SAG or vehicle control for 2 h. Cells were then incubated with VivoGlo™ Luciferin and EnduRen™ In Vivo Renilla Luciferase Substrate (Promega) and immediately assayed with a plate luminometer. Luminescence measurements were taken every 30 s until peak luminescence measurements were recorded.

**Flow cytometry**. NSC GFP fluorescence was analyzed with a BD FACS Aria II flow cytometer. See Supplementary Fig. 15 for FACS gating strategy.

**Statistics and reproducibility**. Statistical analysis was performed using Prism 7 with one- or two-way ANOVA with pairwise Tukey's multiple comparison test for data with multiple groups. A *p* value of <0.05 was deemed to be statistically significant. All values and graphs are shown as mean ± SD. Each experiment used to generate micrographs were independently repeated at least five times, and micrographs of representative samples are presented.

**Reporting summary**. Further information on research design is available in the Nature Research Reporting Summary linked to this article.

## Data availability
The authors declare that all data supporting the findings of this study are available within the article and its Supplementary Information files or from the corresponding author upon reasonable request. *LlSmo* Exon 2 and *L. Lugubris* Sox2 promoter sequences generated in this study have been deposited in GenBank under accession codes MZ983791 and MZ983792, respectively. Source data are provided with this paper.

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

## Acknowledgements

We would like to acknowledge funding from NIH R01GM115444.

## Author contributions

T.P.L., R.L., A.X.S., and M.L.H. performed experiments, analyzed data, and revised and edited the paper. T.P.L. acted as mentor to R.L., A.X.S., and M.L.H.

## Competing interests

The authors declare no competing interests.
