## [Peer Review File · Nature Communications]

Introducing dorsoventral patterning into adult regenerating lizard tails with gene-edited embryonic neural stem cells.Reviewers' Comments:

Reviewer #1:

Remarks to the Author:

In this manuscript Lozito et al determine the molecular signals that inhibit ependymal tube regeneration in an adult lizard. They use CRISPR technology to knock-out the signal inhibitory to correct dorsoventral patterning of the ependymal tube in cultured neurospheres and through transplantation experiments so this suffices to induce ependymal tube regeneration after tail amputation in an adult lizard.

Overall it is a very interesting manuscript that gives novel insights into the roadblocks to regeneration in the lizard and shows how one of these can be overcome but does not give any information on other pathways that may also limit the obstacle of improving neural regeneration at the expense of cartilage regeneration. The manuscript focuses heavily on shh signalling but does not consider the pathways like Fgf, Wnt and BMP that will be established from in vivo work in a range of organisms to play important roles in patterning the neural tube and directing neuronal differentiation.

The manuscript is very dense and in places difficult to read, many figures need better labelling as there are many panels under one label.

Specific points:

1. Figure shows that markers of dorsoventral patterning are expressed during embryogenesis but not during tail regeneration. Pax6 is a marker of lateral tissue, here it appears to be also expressed in the dorsal domain (fig.1 F). Is there overlap in the cells that express Pax6 and Pax7 in the embryos?
2. In fig.1 H, J adult ET shh, a ventral marker appears to be expressed in the dorsal region? Are these panels inverted? Are these cells sox2 positive?
3. Are cells in the ependymal tube in the adult or in the regenerating tail slowly dividing. I would be interested to look at EdU incorporation in these cells.
4. Where are the neuronal cell bodies located in the adult, are there cells which express NeuN and Sox2? NeuN is often found expressed in neural stems that have begun the program of differentiation.
5. Are all cells in the adult or regeneration ependymal tube GFAP and sox2 positive? Are other markers of astrocytes expressed in these cells? Are markers of glial scar formation expressed in the regenerating ependymal tube cells after injury?
6. Do the cells when cultured default to a certain identity because they are missing secreted factors like wnts ?
7. Fig.3 & 4 could be combined into one figure and the text describing them could be shortened.
8. Figure 5 is a nice technique but this could be moved to supplementary data, it seems odd here and disrupts the flow of the manuscript. It is unclear why the authors did not continue to use this technology to label cells for further experiments, it is better than DiI labelling neurospheres.
9. In the neurosphere cultures that authors report they do not see expression of the neuronal marker Tuj1, in the adult derived neurospheres this is a late marker. It is important to determine if the cells begin the neuronal differentiation program but cannot terminally differentiate.
10. The authors use nice clever modern technology to knockout Smo and generate gene edited neurospheres which they implant into the adult tail and assay the outcome of regeneration. The limited data shown suggests that this allows better patterning of the regenerated ependymal tube and differentiation of neurons. What percentage of the regenerated neural tube is derived from the implanted cells? What percentage of cells become neurons? How many animals was this tested on? Do the animals regenerate the same length of tail that was amputated ?

Reviewer #2:

Remarks to the Author:

In this study the authors have performed a detailed analysis of the role of Hedgehog signalling during lizard tail regeneration. They have compared this to developmental roles including a detailed analysis

of neural stem cells. Remarkably they have been able use this knowledge to improve upon tail regeneration by restoring aspects of dorsal/ventral patterning that are normally absent in regenerated tails. The study is also impressive in its development of many seemingly novel techniques for the analysis of regenerative biology in lizards. Over all I would strongly support publication. Here are my comments:

- 1) Figure 1 HJK - Isn't the expression of Hedgehog more widespread than discussed in the results?
- 2) Figure 5 - Aren't there more RFP+ cells than just in the dorsal root ganglion and neural tube? If these are derived from sox2-cre expressing cells shouldn't they also be neurons?
- 3) More quantification is needed. The quantification in the supplemental data is impressive, but more numbers should be provided for each figure. For example N=6 for figure 5 does not provide enough information.
- 4) Some of the figures are very complicated and difficult to follow. There are useful diagrams to show the method, please also add summary diagrams to illustrate your interpretation/model. Doing this for Figure 5 to 8 would be good.
- 5) Please go over the text to improve the logical flow: you should avoid using the word "Next we" to introduce an experiment. Recap and introduce your hypothesis before describing the next experiment. Th authors have done this very well for fig 8/pg 10.
- 6) Consider capitalising "Hedgehog" when referring to the pathway.
- 7) Isn't there also a neural crest contribution to dorsal root ganglion? Shouldn't this be considered in the interpretation of the results?
- 8) It should be pointed out that neural mesodermal progenitors (NMPs) have been identified in other models and these cells are thought to be embryonic tail stem cells capable of generating both neural and mesodermal derivatives. NMPs co-express tbx genes with sox2. It would be good to consider the role of NMPs by including tbx gene analysis in future studies.

Henry Roehl

Reviewer #3:

Remarks to the Author:

This is an exciting manuscript by Lozito et al showing that a loss of dorsal-ventral patterning during tail regeneration in lizards can be re-activated through CRISPR/Cas9 gene editing. This would seem first glance as a niche finding, but it actually has broad implications for regenerative medicine. It is an exciting example of enhancing an in vivo regenerative response in an otherwise limited regenerative capacity. This should have broad interest to researchers working on spinal cord regeneration, but also people investigating non neural regeneration that may use similar strategies to promote regeneration in other organs. It is one of very few examples showing enhancement of regeneration in vivo.

Noteworthy results include the de novo dorsal pattern in an amniote that would otherwise not generate this pattern during spinal cord regeneration. It is of interest that shh expression is the default state of the regenerating tail, which can even overcome embryonic dorsal neural tube cells. I found it particularly striking that the dorsal identity of smoothed KO NSCs generated what looked to be very large DRG structures. To me, this is a significant feat to generate a DRG that otherwise would not be generated. It will be important in the future (not here) to understand whether this DRG innervates the appropriate targets and sends processes back to the spinal cord.

There are some small, unfortunate findings that limit the impact (although very high) such as the observation that the smo KO NSCs do not generate CNS neurons and that a double neural tube is generated after engraftment. Regardless, the manuscript clearly shows that using gene editing can make a NSC competent to participate in proper patterning for spinal cord regeneration, which is an important feat for future therapeutics.

If I was to have one critique on the methodology, the manuscript relies heavily on a single staining modality, IHC instead of incorporating qPCR or FISH as second forms of validation. My concern for this is dampened because the proteins the authors are assaying are known to have specific, robust expression patterns in the neural tube and the staining throughout the manuscript is very specific. Therefore, for this particular case the single staining approach is sufficient.

Overall, this manuscript was a logical and clear. Although the final outcome would have been more impactful if a single neural tube was formed and CNS differentiation occurred after stem cell transplant, this is indeed an exciting advance using clever techniques for tissue grafting and lineage tracing.

Below are some minor comments and mistakes I found in the manuscript:

Page 3, line 13 – Change NCS to NSC

Page 10, line 9-17: Is there any indication why the embryo-derived NSCs/cyclopamine-treated cells don't incorporate into the existing ependymal tube? I would think that it would just generate a dorsal portion of the existing ET. This is quite interesting and may be worth adding a sentence in the results section as it read like an incomplete explanation for what is going on here. Is it because the entire animal is treated with cyclopamine?

In Figure 8H-O, why are most of the GFP+ cells sox2-? Does removing Shh signaling make them differentiate?

Page 12, line 3: I think the phrase "exhibited Col2+ cartilage rods dorsal to ETs" was meant to say ventral instead of dorsal

James Monaghan
Department of Biology
Institute for Chemical Imaging of Living Systems
Northeastern University
Boston, MA 02115

We thank the reviewers for their insightful and thoughtful feedback. This document addresses the concerns and comments that were brought up by the reviewers.

Reviewer #1 (Remarks to the Author):

1. Figure shows that markers of dorsoventral patterning are expressed during embryogenesis but not during tail regeneration. Pax6 is a marker of lateral tissue, here it appears to be also expressed in the dorsal domain (fig.1 F). Is there overlap in the cells that express Pax6 and Pax7 in the embryos?

Thank you for this observation, which is spot on. There appears to be some overlap between Pax7⁺ and Pax6⁺ cells in lizard tail neural tubes, an observation that has been added to the results section of the revised manuscript. We suspect this to be indicative of differences in cell origins and patterning of primary vs secondary neural tubes. Indeed, in embryonic lizard trunks, primary neural tubes (which originate from neural plates) exhibit more distinct lateral Pax6 expression. In contrast, secondary neural tubes of embryonic lizard tails (which originate from tail bud neural mesodermal progenitors) exhibit a dorsal shift in Pax6 expression, as observed in Fig. 1F. We have begun comparing primary vs secondary lizard neural tubes by single cell RNA sequencing to further investigate differences in cell identity and patterning that might affect divergent healing capabilities. This will be the subject of a future manuscript.

2. In fig.1 H, J adult ET shh, a ventral marker appears to be expressed in the dorsal region? Are these panels inverted? Are these cells sox2 positive?

Thank you for your observation. These images are oriented correctly and depict one of the most important findings of this study; adult original and regenerated tail ependyma/NSC populations exhibit floor plate markers such as Shh completely around central canal circumferences, including dorsal regions. Some transverse sections of adult original tail ependyma, such as those depicted in Fig. 1 H, J, exhibit patchier Shh expression than regenerated tail sections, possibly due to the presence of Shh⁺ Tuj⁺ Sox2⁻ neurons. These points have been added to the manuscript text in the revision.

3. Are cells in the ependymal tube in the adult or in the regenerating tail slowly dividing. I would be interesting to look at EdU incorporation in these cells.

We thank the reviewer for their interest, and these EdU/proliferation studies have been previously published by our group (Sun et al. PNAS 2018, Figure S6). For convenience, we included a simplified version of the published figure below. 5-ethynyl-2'-deoxyuridine (EdU) incorporation/staining assays were used to visualize proliferating Sox2⁺ NSC populations in original and regenerating (14, 28, and 56 DPA) lizard tails along their lengths (proximal, middle, and distal). 14-DPA samples, which correspond to the blastema stage of regeneration, exhibited the most numbers of proliferative NSCs. 28-DPA samples, which correspond to lengthening tails, exhibited higher numbers of proliferative NSCs in distal regions. 56-DPA samples, which correspond to fully regenerated tails, resembled original tails and exhibited low levels of proliferating cells.

4. Where are the neuronal cell bodies located in the adult, are there cells which express NeuN and Sox2? NeuN is often found expressed in neural stems that have begun the program of differentiation.

Thank you for the suggestion. To address this comment, we analyzed cross sections of embryonic, adult, and regenerated lizard tails with NeuN, Sox2, and Tuj1 immunofluorescence. NeuN signal was detected in nuclei of Tuj1+ neurons in interior regions of original adult spinal cords, but not in Sox2+ ependymal cells. NeuN signal was not detected in regenerated tail spinal cord, corroborating our observations that no new neurons are formed during tail regrowth. Sox2/NeuN co-expression was only observed in tail neural tube cells, indicating spinal cord neurogenesis was restricted to embryonic stages. This new data is presented in new Supplementary Fig.4.

5. Are all cells in the adult or regeneration ependymal tube GFAP and sox2 positive? Are other markers of astrocytes expressed in these cells? Are markers of glial scar formation expressed in the regenerating ependymal tube cells after injury?

To address this comment, we quantified the percentages of GFAP+ ependymal cells in original and regenerated tails that expressed Sox2. Cross sections of regenerated tail ependymal tubes were immunostained for GFAP and Sox2, and individual cells were identified with DAPI staining. Keyence microscope software was used to quantify the percentages of ependymal tube cells that exhibited GFAP and/or Sox2 signal. Ten separate tails/animals were assayed, and average percentages +/- standard deviations were calculated. 94.3 +/- 3.8% of GFAP+ original tail ependymal cells were positive for GFAP, while 95.2 +/- 4.1% of GFAP+ ET cells were positive for Sox2. This data is presented in new Supplementary Fig. 9.

To determine other markers expressed by regenerated tail NSCs, we referenced our data sets from single cell RNA sequencing data analyses of lizard ependymal tubes and spinal cords and neurospheres, included in the figure below. Regenerated tail GFAP+ Sox2+ NSC clusters co-expressed radial glial markers FABP7 and Sox9. Original tail spinal cord data sets exhibited distinct clusters of GLUL+ astrocytes and Sox10+ MBP+ CLDN11+ oligodendrocytes. These clusters were not represented in ependymal tube or neurosphere data sets, and regenerated tail NSCs did not express other astrocyte and oligodendrocyte markers, confirming their radial glia/ependymal identity. We also did not observe genes commonly associated with mammalian glial scar formation (CSPGs, laminin, tenascin, fibronectin, etc.) in ependymal tube cell transcriptomes. Similarly, we did not observe pro-inflammatory cytokines (CCL2, CCL5, CXCL8 etc.) in regenerated tail spinal cord pericytes. Future work will be aimed at comparing scar-associated genes in regenerating lizard spinal cord cell populations with those expressed during other models of spinal cord injury/healing. We suspect the regenerated tail environment somehow inhibits glial and pericyte scarring processes characteristic of trunk spinal cord injury.

6. Do the cells when cultured default to a certain identity because they are missing secreted factors like wnts ?

We thank the reviewer for their interest and suggestion. To answer this question, we referenced our single cell RNA sequencing data sets to compare growth factor expression in original tail spinal cord, regenerated tail ependymal tube, and cultured neurosphere NSCs. General trends indicated higher Wnt expression (Wnt1, Wnt5a, Wnt11) in ependymal tubes than in original spinal cords and neurospheres. Interestingly, cultured neurospheres expressed high levels of Wnt5b. Conversely, BMP expression (BMP2, BMP3) was higher in original spinal cord and undetectable in ependymal tubes and neurospheres. Finally, we treated cultured neurospheres with Wnt1 and Wnt5a during differentiation assays and did not detect roof plate or neuronal differentiation. In summary, we concluded that environmental factors such as growth factor signaling do not prevent adult lizard tail NSCs from undergoing neurogenesis during tail regrowth. Instead, perhaps NSCs undergo epigenetic changes between embryonic development and adult tail regeneration that inhibit neural differentiation, and future work will explore this topic. We are currently preparing our single cell RNA sequencing results for a separate publication.

7. Fig.3 & 4 could be combined into one figure and the text describing them could be shortened.

Figures 3 and 4 of the original manuscript have been combined into a single figure (Revised Figure 3), and the text has been adjusted and shortened accordingly.

8. Figure 5 is a nice technique but this could be moved to supplementary data, it seems odd here and disrupts the flow of the manuscript. It is unclear why the authors did not continue to use this technology to label cells for further experiments, it is better than Dil labelling neurospheres.

We appreciate the reviewer's suggestions and have moved old Figure 5 to Supplementary Data (Supplementary Fig. 8)

Our choice to use Dil labelling rather than the lentivirus system to trace cell lineages in neurosphere implantation studies was due to the time and supply costs associated with lentivirus production and use coupled with the large number of NSCs needed for implantation. For example, experiments comparing embryonic and adult NSC behavior post-implantation, such as those presented in revised Figure 6), required increased numbers of neurospheres due to limitations in adult NSC propagation in vivo. Unlike embryonic NSCs, adult NSCs exhibited diminished proliferative potential beyond passage 5, meaning that more lizards had to be used to generate the neurospheres required for these experiments. Furthermore, we didn't use the lentiviral system to label Control-CRISPR cells in gene editing experiments in order to provide the most appropriate controls for Smo-CRISPR cells without confounding effects caused by genomic insertion of lentiviral constructs. We had previously reported on Dil labeling as an effective and cost-efficient means for labeling large numbers of lizard neurospheres with minimal effects on NSC behavior and genetics, filling the requirements for the studies included in this manuscript.

9. In the neurosphere cultures that authors report they do not see expression of the neuronal marker Tuj1, in the adult derived neurospheres this is a late marker. It is important to determine if the cells begin the neuronal differentiation program but cannot terminally differentiate.

We thank the reviewer for their insight and suggestion. To address these comments, we analyzed embryonic and adult NSC-derived neurospheres for expression of doublecortin (DCX) and NeuroD2 (early neuronal differentiation markers) by real-time RT PCR. The results of these experiments are presented in revised Supplemental Fig. 11 and are included below for easy referencing. mRNA was collected from either undifferentiated neurospheres or neurospheres cultured under differentiated conditions for 14 days treated with either vehicle control, cyclopamine, or SAG and assayed for Sox2, DCX, NeuroD2, GFAP, and Tuj1 expression. Each experimental condition involved pooled mRNA from 10 neurospheres and was repeated 3 times. Results are presented as fold changes compared to undifferentiated conditions. Results showed that differentiation caused significant increases in DCX, NeuroD2, and Tuj1 in embryonic tail, but not adult tail, NSC-derived neurospheres. These increases were unaffected by cyclopamine treatment, but SAG treatment caused significant decreases in DCX, NeuroD2, and Tuj1 expression. In contrast, adult lizard NSC expression of Sox2, GFAP, DCX, NeuroD2, and Tuj1 expression were unaffected by either differentiation or Hedgehog signaling

modulation. Future work will be aimed at forcing expression of pro-neurogenesis factors like NeuroD2 to attempt initiating neurogenesis in adult lizard NSCs.

10. The authors use nice clever modern technology to knockout Smo and generate gene edited neurospheres which they implant into the adult tail and assay the outcome of regeneration. The limited data shown suggests that this allows better patterning of the regenerated endepydymal tube and differentiation of neurons. What percentage of the regenerated neural tube is derived from the implanted cells? What percentage of cells become neurons? How many animals was this tested on? Do the animals regenerate the same length of tail that was amputated ?

We thank the reviewer for their inquiries. First, we quantified the contribution of implanted embryonic NSCs to adult regenerated tail dorsal and ventral ET regions. These results are presented in the revised manuscript in Supplementary Fig. 16 and are included below for reference. Smo KO NSCs labeled with GFP and control NSCs pre-labeled with Dil were engrafted to dorsal endepydyma of amputated tail spinal cords. Following 28 days of regrowth, regenerated tails were collected and processed for histology. ET cross sections were analyzed by histology/IF/fluorescence microscopy for GFP/Dil signal and Sox2 expression every 1 mm along entire tail lengths. Horizontal lines drawn through the centers of ETs bisected tube images into dorsal and ventral regions. Quantification of dorsal and ventral Sox2 and GFP/Dil signal areas were performed for each cross-section along tail lengths. Percentages of Sox2+ IF signal areas co-expressing GFP/Dil were calculated for dorsal and ventral ET regions and are presented in the histograms below. Dorsal ET regions exhibited significantly higher GFP/Dil signal from labeled exogenous NSC populations than ventral regions, indicating that starting NSC spatial arrangements were maintained along regenerated ET lengths.

Next, we quantified the percentages of exogenous embryonic NCS undergoing neurogenesis following incorporation in adult regenerating tails. These results are presented in the revised manuscript in Supplementary Fig. 17 and are included below for reference. GFP-labeled Smo KO NCSs and control NCSs pre-labeled with Dil engrafted to dorsal spinal cord ependyma cell populations. Following 28 days of regrowth, regenerated tails were collected and processed for histology. ET cross sections were analyzed by IF/fluorescence microscopy for GFP/Dil signal and Tuj1 expression every 1 mm along entire tail lengths. 10 different animals/tails were analyzed for each condition. Tuj1 and GFP/Dil signal areas were quantified for each cross-section, and percentages of Tuj1⁺ IF signal areas co-expressing GFP/Dil are presented in the histograms below. Tuj1⁺ neural cells co-expressed GFP in lizards treated with Smo KO NCSs, indicating significantly higher levels of neurogenesis in Smo KO NCSs than control cells. *, p<0.001 compared to corresponding Control NCS condition for each measurement.

Finally, we compared the tail lengths regenerated by lizards treated with Smo KO NCSs vs Control NCSs vs untreated controls. Both groups of tails regrew to approximately 1.2 cm 28 DPA. Differences in tail lengths were insignificant between conditions, indicating that incorporation of exogenous embryonic NCSs did not affect regenerated tail lengths. The revised text has been updated with these observations. Furthermore, we emphasized the numbers of animals used in each experiment in the revised text and figure legends.

Reviewer #2 (Remarks to the Author):

In this study the authors have performed a detailed analysis of the role of Hedgehog signalling during lizard tail regeneration. They have compared this to developmental roles including a detailed analysis of neural stem cells. Remarkably they have been able use this knowledge to improve upon tail regeneration by restoring aspects of dorsal/ventral patterning that are normally absent in regenerated tails. The study is also impressive in its development of many seemingly novel techniques for the analysis of regenerative biology in lizards. Over all I would strongly support publication. Here are my comments:

1) Figure 1 HJK - Isn't the expression of Hedgehog more widespread than discussed in the results?

We appreciate this note and have also noted that Shh protein is detected among spinal cord nerves surrounding original tail ependyma. The revised results section has been updated to reflect these observations.

2) Figure 5 - Aren't there more RFP+ cells than just in the dorsal root ganglion and neural tube? If these are derived from sox2-cre expressing cells shouldn't they also be neurons?

This is a good question and, to answer it, we re-analyzed the images from our lineage tracing experiments with Keyence microscope software to quantify the percentage overlap of RFP signal with Sox2, GFAP, and TUJ1 signals. These results are presented in revised Supplementary Fig. 9. Overall, they show that the vast majority of RFP⁺ cells (>98%) co-localize with Sox2, GFAP, or TUJ1 regions within either neural tubes, developing and

adult spinal cords or DRG, indicating that most labeled embryonic NSCs differentiate into tail ependymal cells, glia, or neurons. For example, in embryonic samples, the majority of Tuj1+ signal was observed in peri-neural tube regions rather than neural tubes themselves. We hypothesized that these regions were the sites populated by early neurons that would later form tail spinal cord. Interestingly, adult peripheral nerves were only sporadically labeled with RFP using this system, perhaps indicating additional cell sources of tail peripheral nerves. The revised text has been updated with these observations.

3) More quantification is needed. The quantification in the supplemental data is impressive, but more numbers should be provided for each figure. For example N=6 for figure 5 does not provide enough information.

We have taken this suggestion to heart and have quantified immunofluorescence findings presented in Figures 5, 6, 7, 8, and the results are depicted in the revised Figures 4, 6, 8 and in newly added Supplemental Fig. 9, 11, 16, and 17. Specifically, Keyence microscope software was used to quantify percentages of RFP+ or GFP+/Dil+ regions that co-express Sox2, GFAP, TUJ1, Shh, and Pax7. We also provided better descriptions of the animal and sample numbers associated with each experiment.

4) Some of the figures are very complicated and difficult to follow. There are useful diagrams to show the method, please also add summary diagrams to illustrate your interpretation/model. Doing this for Figure 5 to 8 would be good.

Summary diagrams have been added to revised Figures 5 and 7 to illustrate our interpretation of results from lineage tracing and gene-edited NSC transplantation, respectively.

5) Please go over the text to improve the logical flow: you should avoid using the word "Next we" to introduce an experiment. Recap and introduce your hypothesis before describing the next experiment. The authors have done this very well for fig 8/pg 10.

The text has been edited to improve logical flow and to highlight hypothesis-driven experiments.

6) Consider capitalising "Hedgehog" when referring to the pathway.

We have capitalized "Hedgehog" when referring to the pathway throughout the revised text.

7) Isn't there also a neural crest contribution to dorsal root ganglion? Shouldn't this be considered in the interpretation of the results?

We thank the reviewer for this note as it touches upon several interesting aspects of our study considering the unique case of the lizard tail, the only amniote tail that contains both spinal cord and DRG. In the trunks of amniote embryos, folding of Sox2+ neural plates gives rise to lateral and floor domains of primary neural tubes, while neural plate borders pinch off and contribute to neural tube roof plate domains and Sox2- neural crest. Trunk neural crest cells migrate mediolaterally and begin to express Sox2 before forming DRG. The situation is very different in amniote tails, and even more exceptional in lizard tails. Tails are formed from tail buds, which do not have neural plates or neural folds, and hence, no neural crest cells. Instead, Sox2+ neural mesodermal progenitors (NMPs) derived from Sox- tail bud cells directly differentiate into secondary neural tubes. Mammalian secondary neural tubes degenerate before DRG formation during tail development, and adult mammal tails contain neither DRG nor spinal cord. In contrast, lizard secondary neural tubes survive tail development, and lizards are the only amniotes that exhibit both spinal cords and DRG in adult tails. The origins of amniote tail DRG has not been specifically studied, but lineage tracing data presented here suggest that lizard tail DRG derive from Sox2+ tail bud cells that pass through a neural tube intermediate. These observations have been added to the Discussion section of the manuscript, and hopefully future work will expand on this topic.

8) It should be pointed out that neural mesodermal progenitors (NMPs) have been identified in other models and these cells are thought to be embryonic tail stem cells capable of generating both neural and mesodermal derivatives. NMPs co-express tbx genes with sox2. It would be good to consider the role of NMPs by including tbx gene analysis in future studies.

We thank the reviewer for this wonderful observation, and we certainly plan on looking into the relationship among embryonic NMPs, regenerated tail NSCs, and our gene-edited NSCs. In fact, since our initial submission, we have compared these three populations by single cell RNA seq and observed differential TBX18 expression biased to more embryonic cell origins. However, we did not observe the abilities of either native NSCs or transplanted embryonic NSCs to contribute to mesodermal lineages in regenerated lizard tails that has been reported in salamanders. This may mark another difference between salamanders and lizards and may be responsible for their divergent regenerative potentials and will be the subject of future work.

Henry Roehl

Reviewer #3 (Remarks to the Author):

This is an exciting manuscript by Lozito et al showing that a loss of dorsal-ventral patterning during tail regeneration in lizards can be re-activated through CRISPR/Cas9 gene editing. This would seem first glance as a niche finding, but it actually has broad implications for regenerative medicine. It is an exciting example of enhancing an in vivo regenerative response in an otherwise limited regenerative capacity. This should have broad interest to researchers working on spinal cord regeneration, but also people investigating non neural regeneration that may use similar strategies to promote regeneration in other organs. It is one of very few examples showing enhancement of regeneration in vivo.

Noteworthy results include the de novo dorsal pattern in an amniote that would otherwise not generate this pattern during spinal cord regeneration. It is of interest that shh expression is the default state of the regenerating tail, which can even overcome embryonic dorsal neural tube cells. I found it particularly striking that the dorsal identity of smoothened KO NSCs generated what looked to be very large DRG structures. To me, this is a significant feat to generate a DRG that otherwise would not be generated. It will be important in the future (not here) to understand whether this DRG innervates the appropriate targets and sends processes back to the spinal cord.

There are some small, unfortunate findings that limit the impact (although very high) such as the observation that the smo KO NSCs do not generate CNS neurons and that a double neural tube is generated after engraftment. Regardless, the manuscript clearly shows that using gene editing can make a NSC competent to participate in proper patterning for spinal cord regeneration, which is an important feat for future therapeutics.

If I was to have one critique on the methodology, the manuscript relies heavily on a single staining modality, IHC instead of incorporating qPCR or FISH as second forms of validation. My concern for this is dampened because the proteins the authors are assaying are known to have specific, robust expression patterns in the neural tube and the staining throughout the manuscript is very specific. Therefore, for this particular case the single staining approach is sufficient.

Overall, this manuscript was a logical and clear. Although the final outcome would have been more impactful if a single neural tube was formed and CNS differentiation occurred after stem cell transplant, this is indeed an exciting advance using clever techniques for tissue grafting and lineage tracing.

Below are some minor comments and mistakes I found in the manuscript:

Page 3, line 13 – Change NCS to NSC

This mistake has been corrected, thank you for catching it.

Page 10, line 9-17: Is there any indication why the embryo-derived NSCs/cyclopamine-treated cells don't incorporate into the existing ependymal tube? I would think that it would just generate a dorsal portion of the existing ET. This is quite interesting and may be worth adding a sentence in the results

section as it read like an incomplete explanation for what is going on here. Is it because the entire animal is treated with cyclopamine?

We thank the reviewer for identifying this interesting point. We believe that these results indicate a resistance of embryonic NSCs to fully integrate with adult NSC structures when protected from endogenous Hedgehog signaling (yes, because the entire animal is treated with cyclopamine). This explanation has been added to the text of the manuscript. Specifically, we suspect some differences in hedgehog-sensitive cadherin expression between embryonic and adult NSC populations result in their resistance to intermingle. We are currently investigating these possibilities.

In Figure 8H-O, why are most of the GFP+ cells sox2-? Does removing Shh signaling make them differentiate?

We thank the reviewer for their observation. We have quantified the percentages of GFP+ exogenous gene-edited NSCs that co-express Sox2 and other markers following incorporation into regenerated tail ETs, and these results are presented in Figure 7 Z and Supplementary Fig. 16 and 17 in the revised manuscript. These quantifications showed that both GFP-labeled Smo KO NSCs and Dil-labeled Control NSCs persisted among Sox2+ ependyma following incorporation in regenerated tail ETs. However, only Smo KO NSCs were able to differentiate into Sox2- Tuj1+ neurons, supporting the conclusion that removing Shh signaling allows exogenous embryonic NSCs to undergo neurogenesis within adult regenerated tail structures. These points have been added to the text.

Page 12, line 3: I think the phrase “exhibited Col2+ cartilage rods dorsal to ETs” was meant to say ventral instead of dorsal

This mistake has been corrected, thank you for catching it.

**James Monaghan
Department of Biology
Institute for Chemical Imaging of Living Systems
Northeastern University
Boston, MA 02115**

Reviewers' Comments:

Reviewer #1:

Remarks to the Author:

The authors have addressed all reviewers comments and in doing so have improved the manuscript overall.

Reviewer #2:

Remarks to the Author:

The authors have made many revisions to their initial submission and I am happy that my concerns have been addressed. I would recommend publication of this latest version.

Reviewer #3:

Remarks to the Author:

The authors have adequately responded to the concerns raised by me and the other reviewers. I do not have any more concerns with the manuscript.